# Black phosphorus boosts wet-tissue adhesion of composite patches by enhancing water absorption and mechanical properties

Yuanchi Zhang [1], Cairong Li[1], Along Guo[1], Yipei Yang[2], Yangyi Nie[1], Jiaxin Liao[1], Ben Liu[1], Yanmei Zhou[1], Long Li[1], Zhitong Chen [3], Wei Zhang [1], Ling Qin[1,4] & Yuxiao Lai [1,5,6] ✉

Wet-tissue adhesives have long been attractive materials for realizing complicated biomedical functions. However, the hydration film on wet tissues can generate a boundary, forming hydrogen bonds with the adhesives that weaken adhesive strength. Introducing black phosphorus (BP) is believed to enhance the water absorption capacity of tape-type adhesives and effectively eliminate hydration layers between the tissue and adhesive. This study reports a composite patch integrated with BP nanosheets (CPB) for wet-tissue adhesion. The patch's improved water absorption and mechanical properties ensure its immediate and robust adhesion to wet tissues. Various bioapplications of CPB are demonstrated, such as rapid hemostasis (within ~1-2 seconds), monitoring of physical-activity and prevention of tumour-recurrence, all validated via in vivo studies. Given the good practicability, histocompatibility and biodegradability of CPB, the proposed patches hold significant promise for a wide range of biomedical applications.

Tissue adhesives have been extensively developed in biomedical fields due to their ability to halt bleeding, secure biodevices, minimize additional trauma, and alleviate patient pain[1-5]. Given the presence of water and body fluids like blood or sweat in vivo, it is crucial for tissue adhesives achieve robust adhesion under wet conditions[6,7]. However, the inevitable hydration film on wet tissues often acts as a barrier, preventing the direct contact between tissues and adhesives. Additionally, hydrogen bonds (HBs) formed between water molecules and adhesive functional groups can further attenuate the inherent adhesive interaction[8]. Therefore, optimizing wet-tissue adhesion remains a challenge. In nature, organisms like mussel and barnacle have evolved mechanisms to mitigate the detrimental effects of hydration films, thereby achieving good adhesion[9-12]. Inspired by this, Zhao and coworkers introduced a dry-crosslinking tape and a repellent crosslinking paste to reduce the hydration film and ensure wet adhesion[13,14]. These methodologies, termed hydrophilic water-removal and hydrophobic water-removal, predominantly revolve around the mechanical characteristics of the adhesives and the molecular interaction between adhesives and tissues[15]. For example, tape-type adhesives have been proposed to exhibit excellent mechanical properties to obtain robust wet adhesion. A multitude of surface interactions - encompassing hydrogen bonding[16], dynamic covalent bonding[17], bionic adhesion[18,19] and topological adhesion[20], have been introduced to bolster the affinity of tissue adhesives to functional groups, such as NH- groups present on the wet tissue surfaces.

Black phosphorus (BP), a two-dimensional (2D) layer-structured nanomaterial, is notable for its biodegradability and benign degradation products when used judiciously in vivo[21-24]. Of particular interest is

[1]Centre for Translational Medicine Research & Development, Institute of Biomedical and Health Engineering, Shenzhen Institute of Advanced Technology, Chinese Academy of Sciences, Shenzhen, China. [2]Department of Orthopedic Surgery, Shenzhen Hospital, Southern Medical University, Shenzhen, China. [3]Research Center for Biomedical Optics and Molecular Imaging, Institute of Biomedical and Health Engineering, Shenzhen Institute of Advanced Technology, Chinese Academy of Sciences, Shenzhen, China. [4]Musculoskeletal Research Laboratory, Department of Orthopaedics & Traumatology, The Chinese University of Hong Kong, Hong Kong SAR, China. [5]Guangdong Province Engineering Laboratory for Biomedical Materials Additive Manufacturing, Shenzhen, China. [6]The Key Laboratory of Biomedical Imaging Science and System, Chinese Academy of Sciences, Shenzhen, China. ✉e-mail: yx.lai@siat.ac.cn

the degradation of BP nanosheets, which has been recently reported to produce phosphate anions, including $PO_2^{3-}$, $PO_3^{3-}$, and $PO_4^{3-}$, in the presence of illumination, oxygen, and water[25–27]. Generally, the swelling property of materials arises from the interaction between hydrophilic polymer chains and water molecules[28]. Many polymeric networks possess both cationic and anionic groups, either along their main chains or side chains[29–31]. Introducing phosphate anions may rearrange the inter- and intrachain attractions between opposing charges in a polymer system. The dissociation of self-associated polymer chains shall make the matrix swell easily[32]. As a result, the phosphate anions derived from the degradation of BP nanosheets may achieve a strong binding energy with most cationic groups of the polymers, which is expected to enhance the water absorption of tape-type adhesives and further improve the wet adhesion performance. With their high near-infrared (NIR) absorption and efficient photothermal conversion, BP nanomaterials are becoming valuable in photothermal therapy (PTT) for oncological applications[33–36]. Additionally, the inherent electrical conductivity of biodegradable BP nanomaterials has led to their use in crafting conductive hydrogels for tissue engineering[37–39].

In this work, we propose a composite patch integrated with BP nanosheets, referred to as CPB, designed for wet adhesion. As expected in the proposed composite system, the BP nanosheets significantly enhance the water absorption capacity of CPB compared with that of composite patches without BP nanosheets (CP). Owing to the designed triple molecular network and integrated BP nanosheets, CPB retains excellent flexibility and strength, preserving its physical and dynamic attributes post-water absorption. This enables CPB to rapidly adhere to a variety of wet tissues, without interference from blood or other bodily fluids. Thus, CPB exhibits good adhesion strengths, achieving ~171 KPa on wet porcine skin and ~252 KPa on wet nude mouse skin. Leveraging its good water absorption and wet-adhesion capabilities, we demonstrate CPB's potential as a tissue adhesive for various applications. These include rapid hemostasis, physical-activity monitoring, and effective tumour-recurrence prevention, both in vitro and in vivo.

## Results

### Design and mechanism of the composite patch

The bulk of CP and CPB is made of methacrylate anhydride (MA) modified hyaluronic acid (HA) (HAMA), gelatin (Gel), and poly(vinyl alcohol) (PVA) (see Supplementary Methods, Supplementary Fig. 1 to Fig. 3 for preparation details). The synergistic intercrosslinking between HAMA and Gel, coupled with the self-crosslinking of HAMA, affords the material a double-crosslinked structure, thereby enhancing its biomechanical properties. PVA was added to further enhance the mechanical performance. In addition, dopamine-modified polyacrylic acid (PAA-DA) was used for surface treatment, enabling topological entanglement with the molecular chains to form a third crosslinked network around the CPB surface. In the presence of the hydroxyl groups in DA, HBs with NH- groups on tissue surfaces were expected to further improve the adhesive performances of the patches. Figure 1 describes the wet adhesion mechanisms and the potential applications of the proposed patches. Compared to CP, CPB offers enhanced swelling capacity upon adhesion to wet tissues. This mitigates the negative effects of the hydration film on the wet tissues, bolstering the adhesive strength of CPB. For biomedical applications, CPB excels in absorbing excess fluids, promoting blood coagulation, and sealing wounds for rapid hemostasis[40,41]. In addition, the conductivity of the BP nanosheets makes CPB a wearable biosensor, apt for monitoring physical activity in organisms. Last, after surgical tumour resection, CPB can be directly adhered to the surgical site to prevent tumour recurrence by employing the photothermal effect of the BP nanosheets to ablate cancerous tissues under NIR irradiation.

### Patch structure and water absorption capacity

In this work, we first characterized the structure of CP and CPB. With an increase in the content of BP nanosheets (CP: 0 mg, CPB-0.2: 0.2 mg,

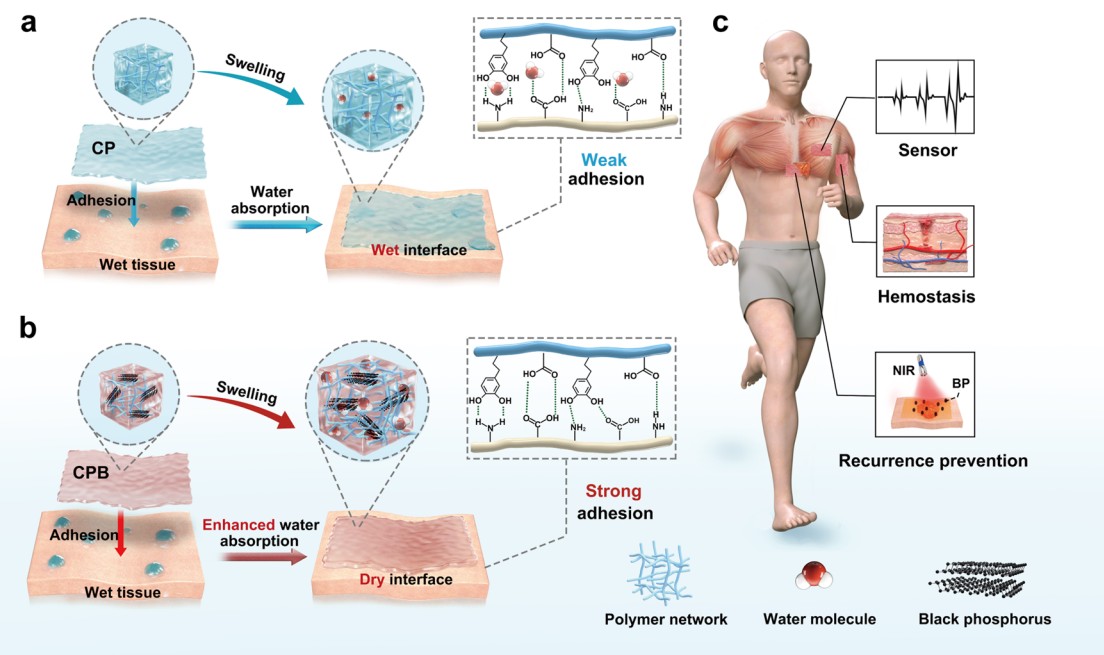

**Fig. 1 | Schematic of the wet-tissue adhesion of the CPB and the potential applications. a** Wet adhesion of CP. With its original swelling ability, CP might not remove all the interfacial water. The interaction between the patch and the tissue is also affected negatively by water molecules, finally resulting in weak adhesion. **b** Wet adhesion of CPB. With a strong swelling capacity when adhered to wet tissues, the interfacial water can be reduced or even removed. The interaction between the patch and the tissue is not affected by water molecules and finally resulting in stronger adhesion of CPB. **c** Potential applications of CPB: rapid hemostasis by promoting blood coagulation and sealing the wound, physical activity biosensor by adhesion to certain sites and tumour recurrence prevention under NIR light through PTT.

CPB-0.6: 0.6 mg, CPB: 1 mg, CPB-1.2: 1.2 mg), the colour of the patches became darker, while CP was white (Fig. 2a). Scanning electron microscopy (SEM) and energy dispersive spectrometry (EDS) images indicated the surface morphologies of the lyophilized CP and CPB series with homogeneously distributed BP nanosheets (Supplementary Fig. 4). Due to their excellent flexibility and stability, the patches could adhere to bent areas and be stored as rolls, ensuring conformal contact with deformed tissue (Fig. 2a, Supplementary Fig. 5a, and Supplementary Movie 1). The typical $A^1_g$, $B^2_g$, and $A^2_g$ models of the BP nanosheets in the Raman spectra and the characteristic bands at 1086, 1024 and 947 $cm^{-1}$ in the FTIR spectra of CPB proved the successful introduction of BP nanosheets (Supplementary Fig. 5b and c). The BP nanosheets can degrade into phosphate anions such as $PO_4^{3-}$ upon contacting oxygen and water[25,26]. CP and CPB contain zwitterionic groups such as $-N^+$ and $-COO^-$ due to the existence of HAMA, Gel and PVA[42–44]. The electrostatic interactions between these cationic and anionic groups contributed to the self-association of polymer chains within the patches. In our hypothesis, the BP nanosheets were stable in

the dried CPB matrix while started to degrade upon contacting the oxygen and water, resulting in the generation of phosphate anions such as $PO_4^{3-}$ and the subsequent dissociation of the originally self-associated polymer chain (Fig. 2b). X-ray photoelectron spectroscopy (XPS) was employed to determine the chemical composition of CP and CPB. The binding energies of nitrogen (N1s) in the CP and CPB spectra for primary, secondary, and tertiary amine peaks were calculated (Supplementary Fig. 5d and Fig. 2c), where the contribution at ~401.4 eV was due to charged amine moieties[45]. The phosphorus binding energies (P2p) showed P−P bonding energies of ~130.1 eV and ~129.1 eV, characteristic of crystalline BP. A strong subband corresponding to the P−O bonding energy was found at ~133.4 eV, indicating that phosphate anions such as $PO_4^{3-}$ were generated owing to the degradation of the BP nanosheets (Fig. 2d)[46].

The swelling capacity of the patches was evaluated by immersing the patches into pure water (Fig. 2e), and by placing the patches into an environmental chamber with 90−95% relative humidity (RH) at room temperature (RT) (Fig. 2f). Figure 2e shows that the patches with BP

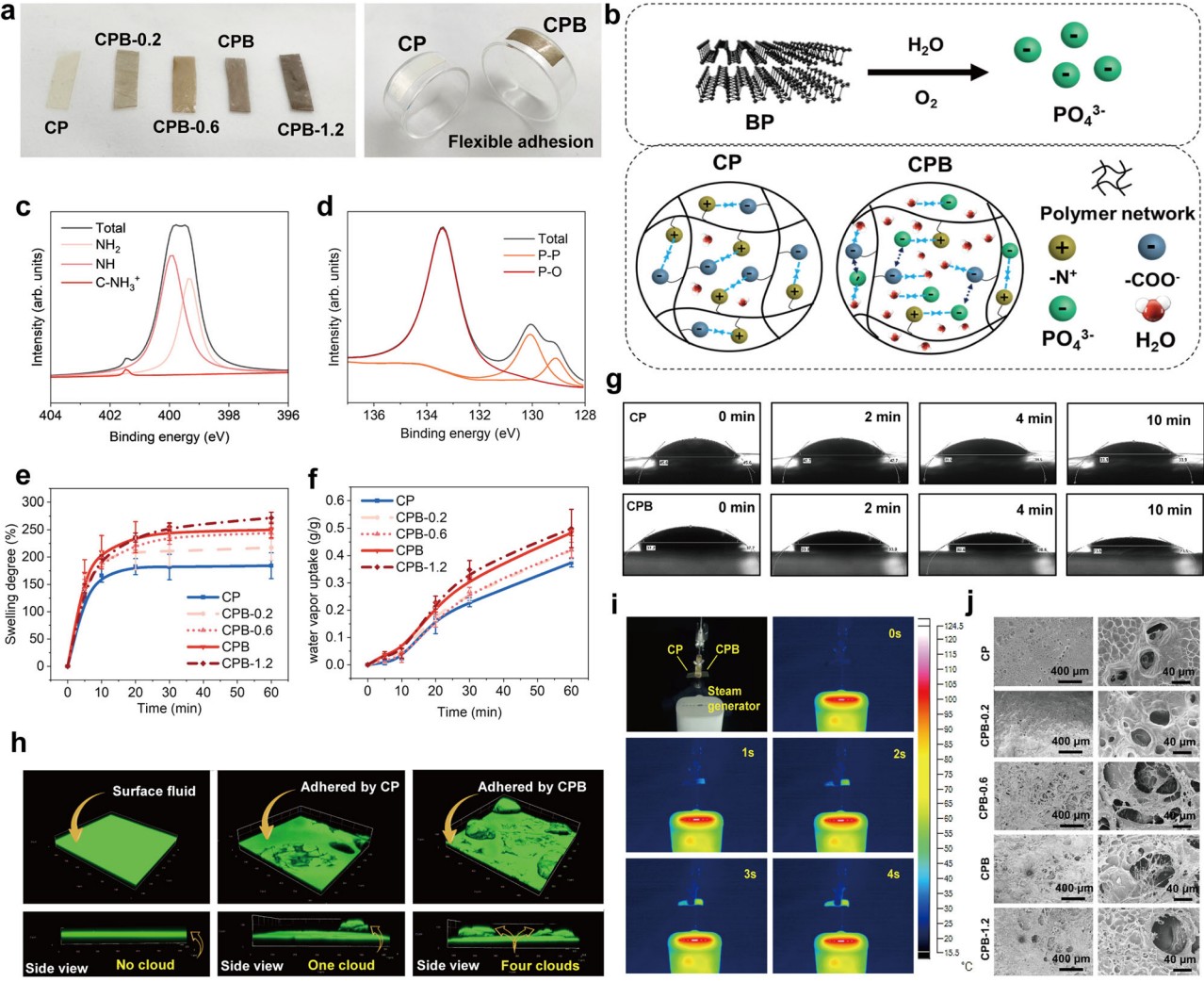

**Fig. 2 | Structures of the patches and the swelling capacity of CPB. a** Images of patches with various contents of BP nanosheets: CP: 0 mg, CPB-0.2: 0.2 mg, CPB-0.6: 0.6 mg, CPB: 1 mg, CPB-1.2: 1.2 mg; flexible adhesion and easy storage of the patches. **b** Schematic illustration of the degradation of the BP nanosheets and proposed structures of CP and CPB. **c** Binding energies if nitrogen (N1s) in the XPS spectrum of CPB. **d** Binding energies of phosphorus (P2p) in the XPS spectrum of CPB. **e** Swelling capacity of CP and CPB in 60 min at RT. **f** Water vapor sorption capacity of CP and CPB in 60 min in an environmental chamber with 90−95% RH at

RT. **g** Contact angles of water drops on the patches in 10 min. **h** Confocal fluorescence microscopy images of the patches before and after being swelled for 1 min. Green, mixture of fluorescent dye (calcein). Cloud refers the fluid being swelled. **i** Infrared thermal images of the patches in hot steam. **j** SEM images (60 x and 500 x) of the patches after being lyophilized. The images were repeated at least twice with consistent results. Values in (**e**) and (**f**) represent the mean and standard deviation ($n = 3$ independent samples).

nanosheets had a greater swelling degree than CP at the initial stage, implying a faster water-swelling speed. After 60 min of immersion, CPB-1.2 and CPB had swelling degrees of ~272% and ~250%, respectively, while that of CP was only ~184%. Even for CPB-0.2, the swelling degree was approximately 220%. A similar trend was observed in the water vapor sorption curves in Fig. 2f, in which more BP nanosheets resulted in the higher water uptake. CPB showed a water vapor sorption capacity of ~0.48 g g$^{-1}$, while CP was only ~0.37 g g$^{-1}$ after 60 min at 90−95% RH. Time-dependent contact angle measurements were performed to observe the wetting properties and the water swelling process of the patches (Fig. 2g).

Although both materials exhibited hydrophilic surfaces, the water contact angles for CPB were consistently lower than those for CP at each time point. After 10 min, the contact angle for CP was ~34°, whereas the water droplet was mostly absorbed by CPB. To visualize the water absorption capacity of the patches, confocal fluorescence microscopy was used with a fluorescent fluid dye (calcein).

As shown in Fig. 2h, the surface fluid absorbed by CP rose to form a singular cloud, while CPB displayed four clouds, clearly representing the better absorption capacity of CPB. Infrared thermal images of the patches (Fig. 2i) further confirmed CPB's rapid water absorption rate. Subsequently, the fully swelled patches were lyophilized, showing that the diameters of pores left by sublimated water became larger with increased BP nanosheets in the patches (Fig. 2j). This might be attributed to the presence of more water in patches with a higher BP nanosheets content, aligning with the previous water swelling experiments.

To elucidate the effect of the BP nanosheets in CPB, density functional theory (DFT) calculations were performed to calculate the interactions between HAMA and Gel segment pairs (① − (a), ① − (b), ② − (a), ② − (b)) and between $PO_4^{3-}$-Gel (a) and $PO_4^{3-}$-Gel (b) pairs (Supplementary Fig. 6). In the absence of $PO_4^{3-}$, −COO$^-$ and O$^-$ groups (① and ②) on the HAMA segments might interact with −N$^+$ groups ((a) and (b)) on the Gel segments through electrostatic interactions. The binding energies of the pairs (① − (a), ② − (a), ① − (b), ② − (b)) were −79.4, −61.8, −90.1, and −60.6 kcal mol$^{-1}$, respectively. On the basis of electrostatic interactions, the polymer chains tended to approach a collapsed state[47]. In the presence of $PO_4^{3-}$, −N$^+$ groups ((a) and (b), especially (a)) were more inclined to combine with $PO_4^{3-}$ because the binding energies $E$ ($PO_4^{3-}$-(a)) and $E$ ($PO_4^{3-}$-(b)) were −111 and −83.5 kcal mol$^{-1}$, respectively, which were much stronger than $E$ (① − (a)), $E$ (② − (a)), and $E$ (② − (b)). Therefore, the phosphate anions derived from the degradation of BP nanosheets might facilitate the disruption of the interactions between the cationic and anionic groups of CPB, resulting in isolated, nonassociated, expanded, and evenly distributed polymer chains that were more easily solvated or more easily absorbed water[31,47].

## Adhesive performance of the patches

On the basis of the enhanced water absorption, CPB was anticipated to exhibit robust adhesive performances on wet tissues, thereby achieving multiple functionalities. Details on the mechanical properties, electrical resistivity, in vitro degradation and photothermal effects of the patches are available in the Supplementary Information. CPB had a Young's modulus of ~102 MPa and a tensile strength of ~11 MPa, which were significantly higher than those of CP, with a Young's modulus of ~45 MPa and a tensile strength of ~6 MPa. Moreover, CPB retained good stretchability with an elongation at break of ~30%, ensuring practical flexibility (Supplementary Fig. 7). These robust characteristics guarantee that CPB sustains both static and dynamic adhesion in wet conditions in vivo, a crucial feature for tissue adhesives. The presence of the BP nanosheets improved the electrical conductivity and sensing sensitivity of the patches (Supplementary Fig. 8). The electrical resistivity of CPB was ~0.35 ohm × cm, closely matching that of Cu foil (~0.33 ohm × cm) and significantly lower than that of CP (~0.70 ohm ×

cm), suggesting good electrical conductivity of CPB. The degradation experiments indicated that, when immersed in PBS at 37 °C, CPB had more than 50% weight loss after one week and more than 75% weight loss after two months (Supplementary Fig. 9), implying good biodegradability of CPB. The faster degradation of CPB might be caused by its improved water-absorption capacity following the introduction of BP nanosheets. As for photothermal effects, the temperature of CPB reached ~54 °C in the wet state after 5 min, a level sufficient to eradicate tumour cells without harming to normal cells[48] (Supplementary Fig. 10).

To assess the adhesive performances of the patches, we carried out two distinct tests: the lap-shear (Fig. 3a) and the modified 180° peel (Fig. 3c) tests, to measure the shear stress and the interfacial toughness, respectively, based on established literature and standards (American Society for Testing and Materials (ASTM) F2256 and ASTM F2255)[49–51]. To evaluate the effects of PAA-DA on the adhesive properties, patches with and without PAA-DA were employed for comparison. Wet porcine organs were first chosen as the model tissues. The adhesion curves can be found in Supplementary Fig. 11. The results indicated that CPB either without or with PAA-DA can establish robust adhesion to wet tissues, which substantially enhances the adhesive strength compared to CP (Fig. 3b, d). For example, the shear stress and interfacial toughness of CPB without PAA-DA adhered to wet porcine skin were ~119 KPa and ~422 N m$^{-1}$, while those of CP without PAA-DA were ~66 KPa and ~378 N m$^{-1}$. This enhancement could be credited to the improved water absorption capacity of CPB. After surface treatment with PAA-DA, the shear stress and interfacial toughness of CPB with PAA-DA adhered to wet porcine skin increased to ~171 KPa and ~638 N m$^{-1}$, compared to that of CP with PAA-DA is ~134 KPa and ~457 N m$^{-1}$, respectively. This was caused by the catechol structure in the PAA-DA molecule, which could improve the interaction between CPB and tissues[52,53]. For further comparison, we used commercial cyanoacrylate tissue adhesives as control groups to evaluate the adhesive effects (Supplementary Fig. 12). As presented, the shear stress and interfacial toughness of commercial adhesive 1 were ~74 KPa and ~73 N m$^{-1}$ for wet porcine skin, while those of commercial adhesive 2 were ~105 KPa and ~88 N m$^{-1}$ for wet porcine skin. CPB presented significantly enhanced adhesion to various wet tissues compared with the commercial adhesives. The improved interaction could also result in a good hemostatic effect.

In addition, the patches were tested on skin tissues harvested from nude mice. The results revealed that CPB (with PAA-DA) showed significant improvements in shear stress (~252 KPa) and interfacial toughness (~251 N m$^{-1}$) compared to CP (~203 KPa and ~144 N m$^{-1}$), respectively (Fig. 3e). Figure 3f proposed a speculative adhesive mechanism of CPB to wet tissues. Upon attaching to wet tissues, the patches first absorbed interfacial fluids such as water and blood to avoid or weaken the negative effects from the water molecular layer. As a result, the patches could quickly form HBs between the tissue surface and then achieve stable strong adhesion. The effects of adhesion time on adhesive strength were also evaluated. The results in Fig. 3g show that the adhesive strength of the patches increased over time. At 0 h, CPB had stronger adhesion (~42 KPa and ~34 N m$^{-1}$) than CP (~27 KPa and ~11 N m$^{-1}$) and increased more than 4 times after 48 h adhesion, underscoring the practicability of CPB for biomedical applications. The adhesive performance of the patches adhered to dry porcine skin after 12 h is shown in Supplementary Fig. 13, indicating that the enhanced water absorption capacity rather than the improved mechanical properties, contributed more to the robust wet adhesion of CPB. The instant adhesion of CPB to various wet organs and blood-covered bones can be found in Fig. 3h, i. To further observe the adhesive performance of the CPB, an ex vivo rabbit stomach model of fluid leakage was prepared, in which CPB successfully sealed a water-filled perforated stomach within a short period, further attesting to the instant adhesive ability of CPB (Fig. 3j and Supplementary Movie 2).

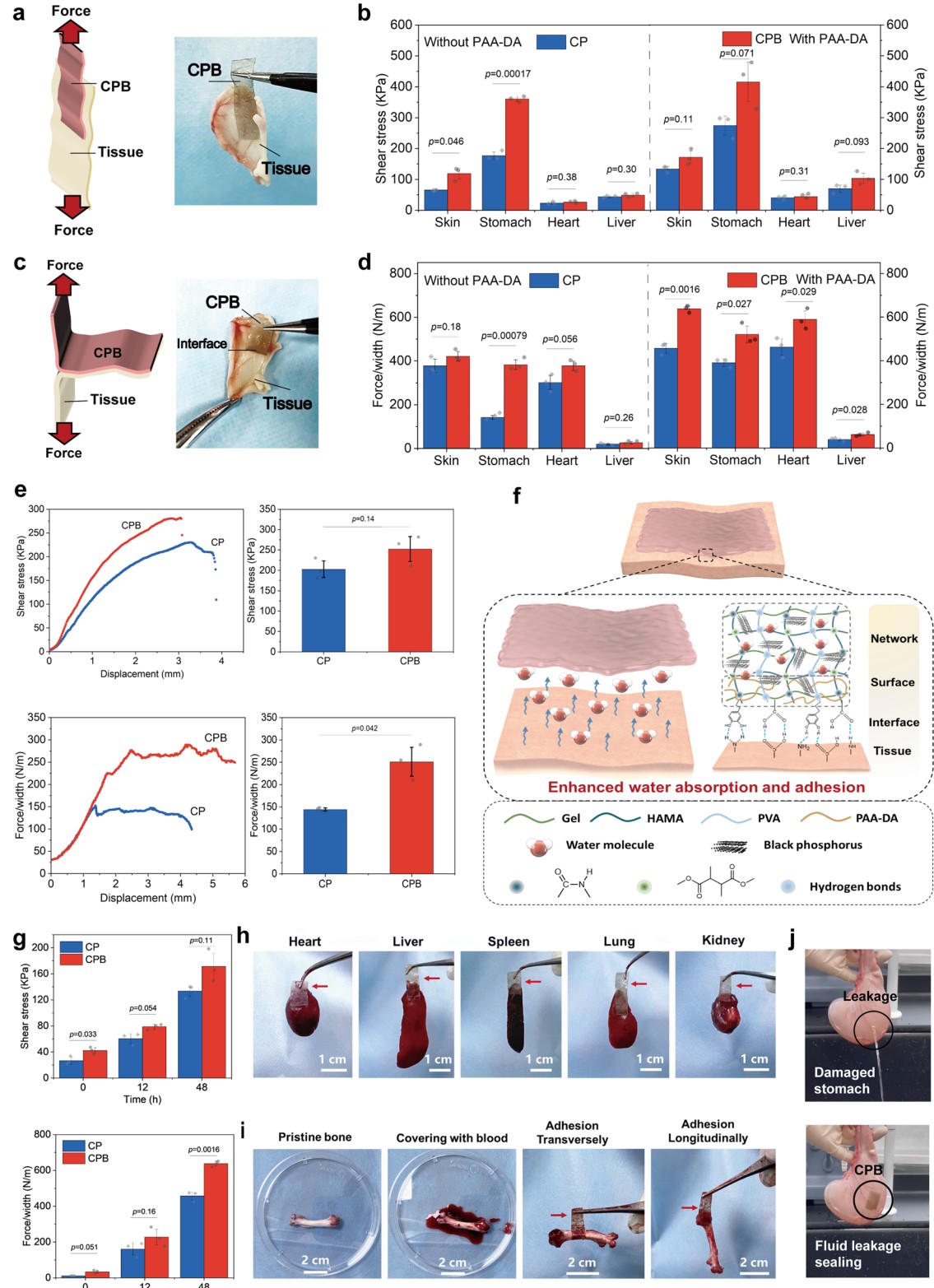

## Potential applications

CPB has great potential for various biomedical applications. In particular, the excellent physical properties and biodegradability of CPB are favourable for short-term applications in vivo. Furthermore, good biocompatibility of the patches was also confirmed (Supplementary Fig. 14). First, we used a normal SD rat liver perforation wound model to investigate the hemostatic effect of CPB (Fig. 4a and Supplementary Movie 3). CP and commercial products (e.g., Gelatin Sponge and

Gauze) were used as the control groups (Supplementary Fig. 15). After perforation, the sample was directly adhered to the wound, while no treatment was applied in the blank group. For the CPB group, only a small area of bloodstain appeared on the surface of the filter paper beneath the liver upon patch adhesion to the wound (approximately 1–2 s) (Fig. 4b). For the CP and Gelatin Sponge groups, the area of bloodstain was slightly greater compared to that in the CPB group. In contrast, a clear bleeding pathway was observed in the Gauze group

**Fig. 3 | Adhesive performances. a** Schematic representation (left) of the lap-shear measurement process and photograph (right) of CPB adhered to tissue for this measurement. **b** Shear stress of CP and CPB (with or without PAA-DA) adhered to porcine tissues from various organs (skin, heart, stomach, liver). **c** Schematic representation (left) of the modified 180° peel measurement process and photograph (right) of CPB adhered to tissue for this measurement. **d** Interfacial toughness of CP and CPB (with or without PAA-DA) adhered to porcine tissues from various organs (skin, heart, stomach, liver). **e** Lap-shear (top) and modified 180° peel (bottom) adhesive performances of CP and CPB (with PAA-DA) adhered to skin tissues from nude mice. **f** Schematic illustration of the enhanced adhesion mechanism of CPB to wet tissues. **g** Effect of adhesion time on the adhesive performance of the patches. **h, i** Images of CPB adhered to a series of representative tissues (**h**) and to bone tissue coated with blood (**i**). Red arrows: CPB. **j** Sealing of a fluid-leaking ex vivo rabbit stomach by CPB. Values in (**b**), (**d**), (**e**), and (**g**) represent the mean and standard deviation ($n = 3$ independent samples). Statistical analyses were performed by using two-tailed Student's $t$-test. No adjustments were made for multiple comparisons. $P$ values less than 0.05 were considered statistically significant differences between the compared groups.

and the blank group (Supplementary Fig. 15a, Fig. 4b). Over time, the bloodstains in the Gauze group and blank group became more obvious, but there was minimal change in the CPB group (Supplementary Fig. 15a, b, Fig. 4b, c). Quantitatively, the total blood loss of the CPB group (-0.07 g) was much less than the blank group (-0.4 g) and the commercial product groups (-0.11 to 0.13 g) (Supplementary Fig. 15c, Fig. 4d). Additionally, a dynamic heart perforation wound model was used, where CPB was rapidly adhered to the wound and then pressed for 3-5 seconds after perforation (Fig. 4e). The hemostatic effect of CP and the commercial products in the control groups can be found in Supplementary Fig. 15d. Once the sample was positioned, no further obvious blood was observed around the wound in the Gelatin Sponge group, the CP group and the CPB group, but it did in the Gauze group. Even acting on a dynamic and curved surface, CPB still exhibited a swift hemostatic effect, which can be attributed to its strong adhesion ability reducing blood diffusion and strengthening the wound seal.

With its conductive properties, CPB can be directly adhered to the skin of a nude mouse and used as a physical activity monitor (Fig. 4f). Figure 4g illustrates the electromyographic (EMG) signals acquired by CP, CPB and Cu foil of the nude mouse from under anesthetic to wake up state. CPB and Cu foil presented the similar level of sensing capability, which was better than that of the CP sample. The physiological activities of wake-up processes could be clearly identified through the amplitude and frequency of the sensing potentials. The presence of BP nanosheets improved the electrical conductivity and sensitivity of the patches, allowing CPB to potentially be popularized as a biomedical sensor.

An additional application of CPB is in tumour recurrence prevention. Tumour cells could be killed in vitro by photothermal treatment (PTT) with CPB, and NIR light was proven to be friendly to normal cells (Supplementary Fig. 14). The enhanced bioadhesion ensured that potential tumour cells remained in a fixed position to receive NIR light. The agenda of the treatment using a nude mouse model is depicted in Fig. 5a. The tumours were removed when their sizes grew to -200 mm³, and then the patches were adhered to the surgical site with or without NIR irradiation (Supplementary Fig. 16). The nude mice in the blank group were sacrificed when the tumour size reached -1933 cm³ at 3 weeks (Supplementary Fig. 17).

Figure 5b presents the thermal images (left) and temperature elevation (right) at the surgical site with CPB under NIR light (808 nm, 1 W cm⁻²) for 5 min. The temperature increased to -50 °C in 60 s and eventually reached -55 °C, confirming the photothermal effect of CPB in a living body. After surgical removal of the tumours, the body weight of the nude mice in each group had an increasing trend but no significant difference, suggesting no serious side effects in this therapy strategy (Fig. 5c). The tumour volumes were calculated according to the width and length every two days (Fig. 5d). Compared with the S group and the S + CP group, tumours in the S + CPB group were smaller, which might imply a certain antitumour effect of BP nanosheets[54,55]. Tumour recurrence in the NIR group was similar to that in the S group, indicating that NIR light alone did not inhibit tumour growth. More importantly, the tumours in the S + CPB + NIR group did not reemerge within 4 weeks, demonstrating the

effective in situ PTT of CPB. Because the MCF-7 cells used for the tumour model were labeled with red fluorescence protein (RFP), the tumour growth process could be monitored by an in vivo imaging system (IVIS). The fluorescence images revealed tumour progression variations across different treatments over 4 weeks (Fig. 5e). Except for the S + CPB + NIR group, all groups had different degrees of tumour recurrence, corroborating the observed tumour volume changes. To evaluate the in vivo toxicity of the patches, we collected major organs from the nude mice in each group after 4 weeks and used a hematoxylin and eosin (H&E) staining assay for histology analysis. No discernible differences were observed between normal tissues and those from the various experimental cohorts, suggesting the good histocompatibility of CPB treatment (Fig. 5f). These results demonstrated that the CPB therapeutic approach possessed effective in situ antitumour functions for postsurgical treatment. In addition, degradation tests of patches with and without NIR light in vivo indicated that CPB could degrade to -40–42% residual amounts in 8 weeks, with NIR light exerting no significant impact on biodegradation (Supplementary Fig. 18). With the degradation of the patches, histocompatibility investigation at 2, 4 and 8 weeks confirmed that the patches, irrespective of NIR light exposure, did not cause inflammation surrounding the implants, suggesting commendable histocompatibility and biosafety of the proposed patches (Supplementary Fig. 19).

## Discussion
In this work, the BP nanosheets were leveraged to prepare CPBs with enhanced water-absorption capacity for wet-tissue adhesion. DFT calculations suggested that phosphate anions from the BP nanosheets facilitated the elimination of interactions between the cationic and anionic groups of CPB. Due to the enhanced water absorption capacity and surface treatment with PAA-DA, CPB can adhere to various wet tissue surfaces and then achieve robust adhesion (i.e., adhesive strengths of -171 KPa and -252 KPa to wet nude mouse skin and wet porcine skin, respectively). In addition, CPB possesses good flexibility and mechanical properties with a Young's modulus of -102 MPa and elongation at break of -30%, which ensures adhesive stability and permits deformation when CPB adheres to active tissues in wet conditions. With the presence of the BP nanosheets, the patches exhibit consistent photothermal effects under NIR irradiation in both dry and wet states and possess low electrical resistivity. Based on the multifunctional properties of CPB, we focused on its short-term bioapplications in this work. CPB was proven to be capable of rapid hemostasis (within -1 to 2 s), physiological activity monitoring, and effective prevention of tumour recurrence in animal models. The cell biocompatibility, biodegradation, and histocompatibility of CPB were also demonstrated.

## Methods
### Ethics statement
All animal experiments were approved by the Institutional Animal Care and Use Committee (IACUC), Shenzhen Institutes of Advanced Technology (SIAT), Chinese Academy of Sciences (SIAT-IACUC-211010-YGS-ZYC-A2067) and performed following the guidelines

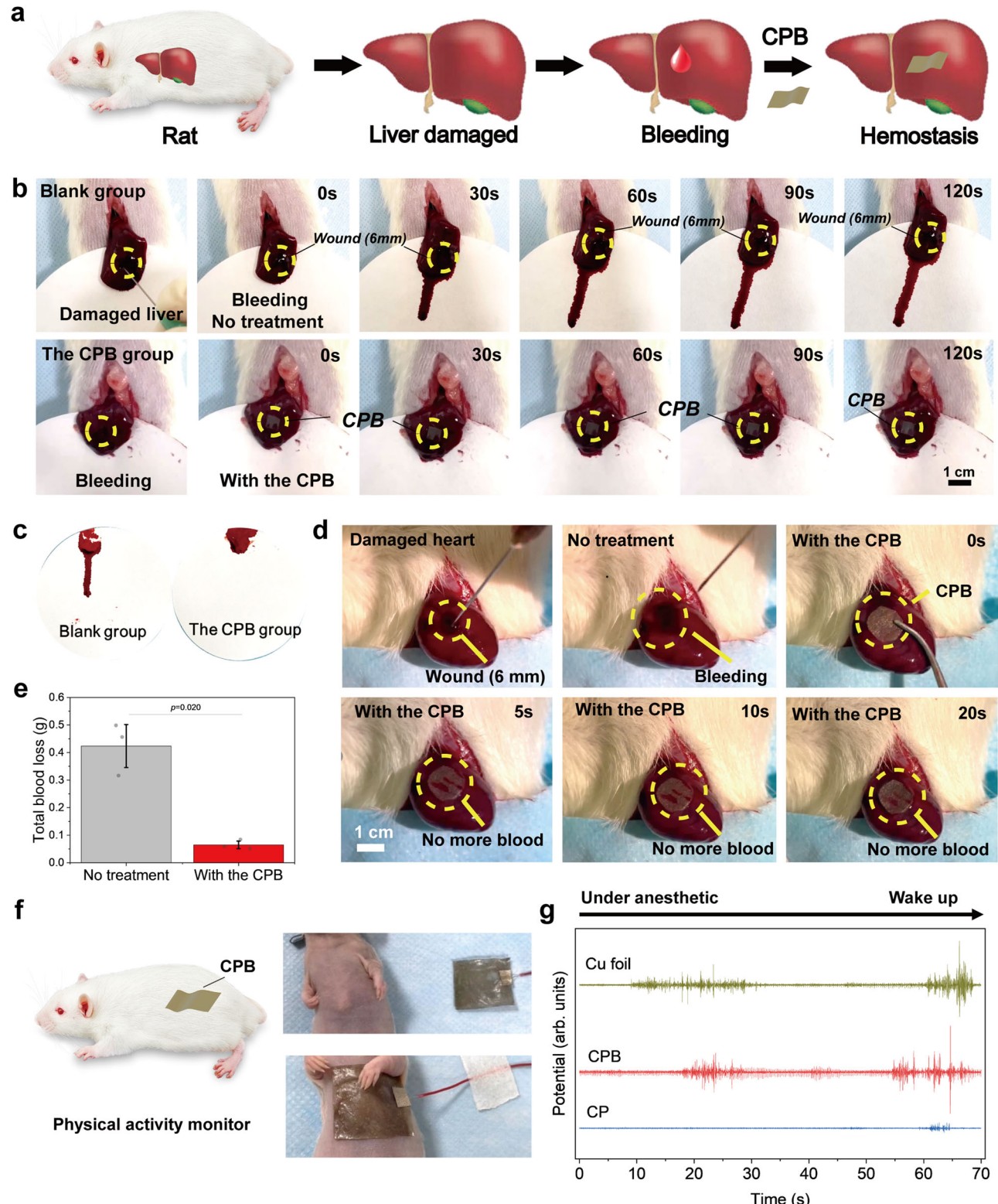

**Fig. 4 | Potential applications in in vivo hemostasis and activity monitoring.**
**a** Schematic illustration of the hemostatic process using CPB in a rat liver perforation wound model. **b** Images of the hemostatic effect on damaged livers in the blank group and the CPB group. Yellow dotted line: position of the wound or CPB.
**c** Bloodstain on the surface of filter paper in the blank group and the CPB group at 120 s. **d** Total blood loss in the blank group and the CPB group. **e** Images of the hemostatic effect of CPB in a rat dynamic heart perforation wound model. Yellow

dotted line: the position of the wound or the CPB. **f** Schematic illustration of CPB as a physical activity monitor and the adhesion of the CPB monitor on a nude mouse. **g** Potentials of the nude mouse from under anesthetic to wake up state. Values in (**e**) represent the mean and standard deviation ($n = 3$ independent samples). Statistical analyses were performed by using two-tailed Student's $t$-test. No adjustments were made for multiple comparisons. $P$ values less than 0.05 were considered statistically significant differences between the compared groups.

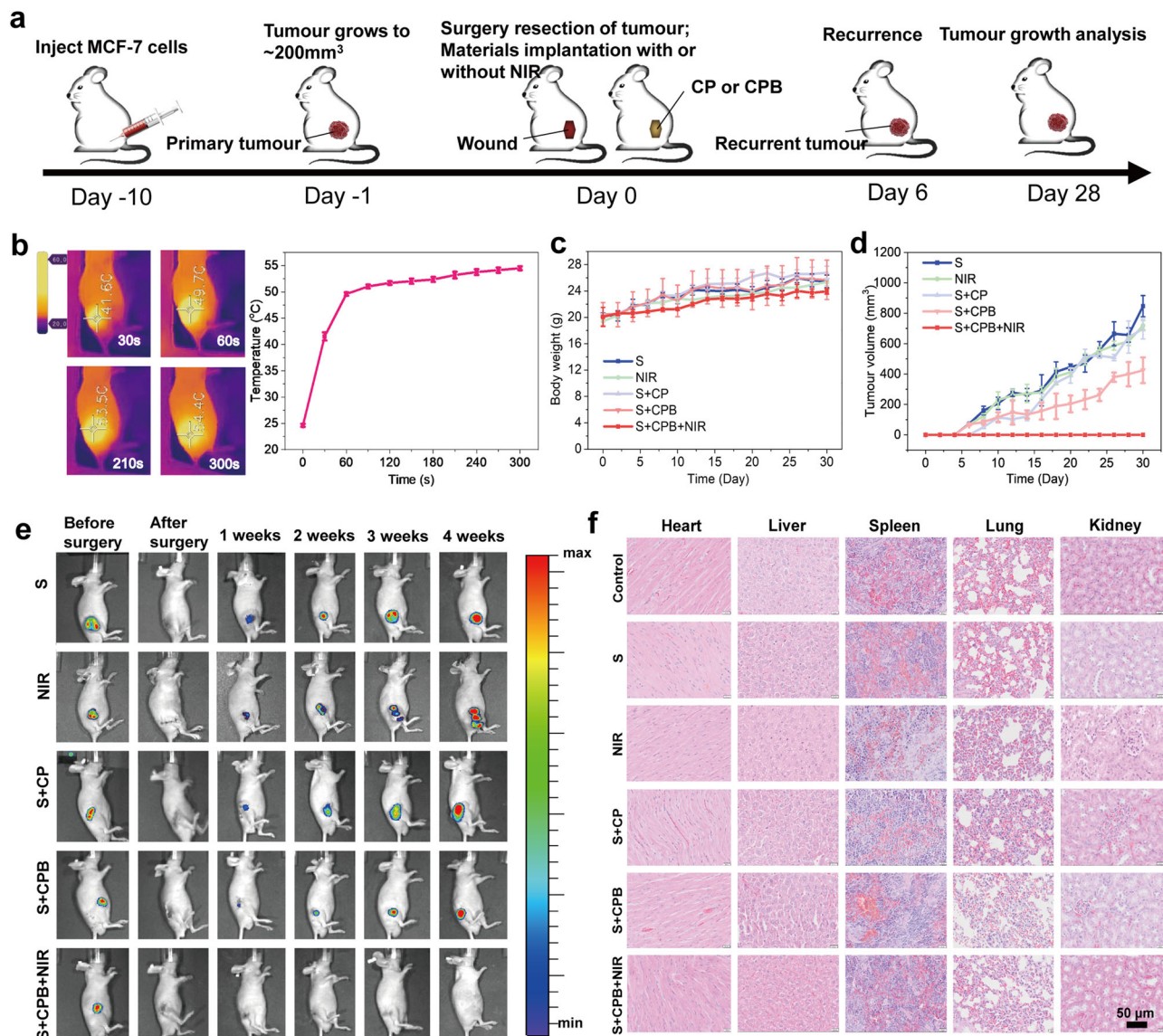

**Fig. 5 | In vivo studies of tumour postsurgical treatment. a** Treatment schedule of tumour inoculation, resection and patch implantation. **b** Thermal images (left) and temperature changes (right) at the surgical site of tumour-bearing mice with CPB under NIR light (808 nm, 1 W cm⁻²). **c** Body weight changes of the nude mice recorded every two days in various groups. S: only surgical resection; NIR: after surgical resection, the surgical site was exposed to NIR light (808 nm, 1 W cm⁻²) for 5 min; S + CP: after surgical resection, CP was adhered to the surgical site; S + CPB: after surgical resection, CPB was adhered to the surgical site without NIR light; S + CPB + NIR: after surgical resection, CPB was adhered to the surgical site subsequently with NIR light (808 nm, 1 W cm⁻²). **d** Tumour volume after various treatments recorded every two days in various groups. **e** Fluorescence images of the tumour-bearing mice with various treatments immediately before and after surgery, and at 1 week, 2 weeks, 3 weeks, and 4 weeks. **f** H&E-stained sections of major organs from the sacrificed nude mice in each group after 4 weeks. The histological investigations were repeated at least twice for consistent results. Values in (**b**), (**c**) and (**d**) represent the mean and standard deviation ($n = 3$ independent samples).

published by the NIH[56]. Experimental animals were housed in specialized cages with six mice per cage and three rats per cage. Alternate 12–14 h of light and 12–10 h of darkness every day. Temperature was set at 20–26 °C with a 30–70% relative humidity. Adequate food and pure water were guaranteed, and clean cages were replaced weekly. The animal was anesthetized, and placed on a thermostatic pad, there was no pain reflex from pinching the toes, and the animal was breathing steadily, which represented the possibility of surgical operation. Penicillin injections were administered for two consecutive days post-surgery to avoid wound inflammation. In accordance with IACUC of SIAT and the NIH guidelines, the maximal tumour size permitted was <2000 cm³, and/or limited within 20 mm in any one dimension. Animals were euthanized if they were too weak to take food and water by themselves, and/or suffered from serious organ infection, and/or their body weight loss > 15%, and/or tumour size > 2000 cm³.

## Material

HA (Mw-74000) was obtained from Bloomage Biotechnology Co., Ltd (Beijing, China). MA, DA, N-hydroxysuccinimide (NHS), gel from porcine skin, PVA (Mw-9000-10000) and 2-hydroxy-4'-(2-hydroxyethoxy)-2-methylpropiophenone (I2959) were purchased from Sigma–Aldrich (Shanghai, China). PAA (Mw-5000, 50 wt%) was purchased from J&K Scientific Ltd. (Beijing, China). N-(3-Dimethylaminopropyl)-N'-ethylcarbodiimide hydrochloride (EDC) was obtained from Aladdin Bio-Chem Technology (Shanghai, China). Commercial tissue adhesives 1 and 2 were purchased from Beijing Compont Medical Equipment Co., Ltd. (Beijing, China) and Zhejiang Perfectseal New

Material Technology Co., Ltd. (Ningbo, China), respectively. The commercial gelatin sponge and gauze were obtained from Wuhan Hongda Co., Ltd. (Wuhan, China). BP nanosheets (1 mg ml⁻¹) with a size of approximately 100 nm · 1 μm and thicknesses of 10–100 nm were procured from Mophos Co., Ltd. (Shenzhen, China). The commercial fresh porcine tissues were purchased from the local supermarket, and then the surface of the tissues was cleaned with saline solution and sterilized by 75 v v⁻¹% ethyl alcohol before the experiments.

## Preparation of CPB
A total of 1 ml HAMA water solution (6 w v⁻¹%), 1 ml PVA water solution (30 w v⁻¹%), 1 ml BP (1 mg ml⁻¹) water solution, EDC (60.5 mg), NHS (36.4 mg) and I2959 (12 mg) were mixed with stirring until the solution became transparent. Then, 1 ml Gel water solution (30 w v⁻¹%) was added to the mixture with fast stirring followed by being poured into a glass mold. Intercrosslinking bonds between HAMA and Gel formed due to the presence of the EDC/NHS linker. Subsequently, the mixture was placed under UV light (365 nm, ‐68 mW cm⁻²) for 5 min, resulting in the self-crosslinking of HAMA. Next, the sample was dried in a fume cupboard overnight at RT to obtain the initial CPB product. According to the literature, surface treatment by allowing PAA-DA to permeate the sample for 1 min could further improve the adhesive strength[52]. Thus, PAA-DA water solution (5 wt%) was dropwise added on the surface of the dried film with a density of 0.5 μl mm⁻², where it permeated into the film to form topological entanglement. Finally, CPB with a triple-crosslinked network was prepared.

## Characterization of the patches
XPS was performed to analyze the protonated and deprotonated forms of the amine-containing and phosphorus-containing chemicals in CP and CPB by an XPS spectrometer (Thermo Scientific Nexsa, USA) using an Al Kα ($\lambda = 0.83$ nm, $hv = 1486.6$ eV) X-ray source operated at 72 W. The swelling degrees of CP and CPB were investigated by dipping the samples in deionized water for various periods. The weights of the samples before and after being dipped at different times were noted for calculation according to the Eq. (1)[57].

$$\text{Swelling degree} = \frac{w_1 - w_0}{w_0} \qquad (1)$$

where $w_0$ (g) is the weight of the original patch and $w_1$ (g) is the weight of the swollen patch at 0, 5, 10, 20, 30 and 60 min, respectively. The water uptake of CP and CPB was measured by placing the patches in a box at an RH of 90‐95% for 60 min to evaluate the water absorption of the patches. Water uptake was calculated by the following Eq. (2).

$$\text{Water uptakes} = \frac{m_1 - m_0}{m_0} \qquad (2)$$

where $m_0$ (g) is the original weight of the patch and $m_1$ (g) is the weight of the patch after being placed in a box at an RH of 90–95% for 0, 5, 10, 20, 30 and 60 min.

The contact angles of CP and CPB were tested by a contact angle meter (Jinhe JY-PHb, Chengde, China). Deionized water was dropped from a needle onto the surface of the patches, and the contact angle was recorded every 2 min. To visually investigate the absorbed water, 5 μl calcein was added to a culture dish, and then the patches were placed on the dye. After 1 min, confocal fluorescence microscopy (ZEISS LSM 900, Germany) was used to observe the patches absorbing the fluorescent fluid. In addition, the dried patches were fixed on an iron support, and a steamer was activated to generate hot water vapor (Fig. 2i). Upon turning on the steamer, infrared thermal images of the patches were taken by an NIR camera (FLIR One, FLIR Systems, Inc., Hong Kong, China) to observe the water uptake of CP and CPB. The

morphologies of the patches after being lyophilized were investigated by SEM (ZEISS SUPRA® 55, Carl Zeiss, Germany) at an accelerating voltage of 5 kV.

## DFT calculations
Density functional theory calculations were carried out using Gaussian 16. Revision C. 01. Geometric optimizations were performed at the M06-2X-D3(0)/6-311 + G(d) theoretical level. Harmonic vibration frequency calculations were performed for all stationary points to confirm them as a local minimum (Nimag = 0). The molecular structures were drawn using CYLview 1.0 b software. Binding energies were calculated as $E_{bind} = E_{a\text{-}b} - E_a - E_b$

## Adhesive properties
To characterize the adhesive performance, the samples were cut to a size of 4 mm (width) × 15 mm (length). To investigate the adhesive properties, one side of the CPB sample was adhered to the harvested tissues followed by gentle pressing with the fingers. CP samples and commercial tissue adhesives were used as the control groups. Then, the modified lap-shear adhesion test and 180° peel adhesion test based on the literature and relevant standards (ASTM F2255 and ASTM F2256, respectively)[49–51] (Fig. 3) were conducted after 0, 12, and 48 h with an Instron tester (Instron Electropuls E10000, USA) at a testing rate of 50 mm min⁻¹ at RT. The force was recorded, and then calculations were performed using the equations (3) and (4): Shear stress = $F_{max}/WL$ (3), where $F_{max}$ is the maximum force in lap-shear test, $W$ and $L$ represent the length and width of the sample, respectively; Toughness = $F/W$ (4), where $F$ is the plateau force in peel test, $W$ and $L$ represent the length and width of the sample, respectively. CPB was adhered to a series of representative tissues (heart, liver, spleen, lung, kidney, bone) harvested from the rats to observe the adhesive performance.

## In vivo studies
In vivo studies of the CPB (1 ml HAMA water solution (6 w v⁻¹%), 1 ml PVA water solution (30 w v⁻¹%), 1 ml BP (1 mg ml⁻¹) water solution, 1 ml Gel water solution (30 w v⁻¹%)) with PAA-DA were conducted.

a. The hemostatic effect of CPB was first evaluated in a normal SD rat liver perforation wound model. Bleeding without treatment was used as the blank group. Gauze and gelatin sponge treatments were used as the control groups. A total of 10 SD rats (male, weight of 250–300 g, 7–8 weeks) were randomly divided into the CPB group and the blank group. Then, the livers of the rats were lifted and placed on the surface of preweighted filter paper, and a circular perforation wound (diameter of 6 mm) was created for hemorrhage. CPB was cut to a size of 5 × 5 × 0.2 mm and weighed in advance. Next, CPB was directly adhered to the bleeding site. The hemostatic process was recorded with a digital camera. The blood loss was calculated by determining the total weight of the blood absorbed by the filter paper and by CPB. In addition, the hemostatic effect of CPB was then evaluated in a normal SD rat heart perforation wound model. Similarly, the hearts of the rats were lifted, and then a circular perforation wound (diameter of 6 mm) was created for hemorrhage. CPB (8 × 8 × 0.2 mm) was immediately adhered to the bleeding site, and the state was recorded with a digital camera.

b. Surface electromyographic (sEMG) signal recording was performed by a specific EMG signal acquisition instrument with Arduino circuit boards (Xinweilai, China), which was able to measure the biological signals. A piece of CPB (25 × 25 × 0.2 mm), serving as EMG sensing electrode, was placed on the skin of the belly of a nude mouse, and the reference electrode was placed on the skin of the tail. CP and Cu foil were used as the control groups. The measured sEMG signals were simultaneously recorded using the MATLAB Platform (Mathworks, Natick, MA). The physical activity of the nude mouse could be monitored in real time.

c. The nude mice were subcutaneously injected with 100 μl cell culture medium at a density of $5 \times 10^4$ cells/μl. The MCF-7 cells used for the tumour model were labelled with RFP; therefore, the tumour growth process could be monitored by IVIS. Samples were cut to a size of $20 \times 10 \times 0.2$ mm. A total of 36 mice were randomly divided into six groups for investigating histocompatibility and tumour recurrence prevention: blank group (no surgical treatment), surgery group (S group: only surgical removal of the tumour, no materials), surgery with NIR light only (NIR: after surgical resection, the surgical site was exposed to NIR light (808 nm, 1 W cm$^{-2}$) for 5 min), surgery with the CP group (S + CP group: after surgical removal of the tumour, CP was placed on the surgical site); surgery with the CPB group (S + CPB group: after surgical removal of the tumour, CPB was placed on the surgical site without NIR light), and surgery with the CPB group with NIR light (S + CPB + NIR group: after surgical removal of the tumour, CPB was placed on the surgical site with NIR light (808 nm, 1 W cm$^{-2}$) for 5 min). During irradiation, an NIR camera (FLIR One, FLIR Systems, Inc., Hong Kong, China) was utilized to monitor the temperature changes of the tumour sites. When the tumour volume reached ~200 mm$^3$, tumours in all groups, except the blank group, were dissected. Different treatments were utilized in those groups to observe the local recurrence of the tumour to investigate the therapeutic effects of CPB. The weights of the mice and the volume of the tumours calculated by the width and the length were recorded every two days until 4 weeks. Then the nude mice were killed, and the tumours were collected. The mice in the blank group were sacrificed at ~3 weeks due to the animal experimental endpoint. In addition, IVIS (PerkinElmer, USA) was used to take in vivo images of the mice every week to monitor tumour growth. A 535 nm wavelength light was selected as the excitation source, and 600 nm was used as the emitted light wavelength. H&E (Sigma-Aldrich, Shanghai, China) were used after 4 weeks to stain the major organs of the mice for histological evaluation. After staining, the sections of the organs were observed by a light microscope (Olympus IX71, Japan).

### Statistical analysis

Two-tailed Student's *t*-test were utilized for statistical analysis, and *P* values less than 0.05 were considered statistically significant differences between the compared groups. Unless otherwise specified, all values the mean and standard deviation (*n* = 3 independent samples). Quantitative results were analyzed using the software SPSS 27.0 and Microsoft Excel 2013. The graphics of the data analysis were prepared though with software Origin 2022.

### Reporting summary

Further information on research design is available in the Nature Portfolio Reporting Summary linked to this article.

## Data availability

All data are available in the main text and Supplementary Information. Source data are provided with this paper. The data generated in this study have been deposited in the figshare database under the accession code https://doi.org/10.6084/m9.figshare.24548476. Source data are provided with this paper.

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

## Acknowledgements

This work was supported by the National Natural Science Foundation of China (82022045 (Y.L.), 52203204 (YC.Z.)); Shenzhen Medical Research Funds (B2302050) (Y.L.); the CAS Interdisciplinary Innovation Team (JCTD-2020-19) (Y.L.); the Shenzhen Basic Research General Project (JCYJ20220531100408019) (YC.Z.); Shenzhen Science and Technology Program (KJZD20230923114612025) (Y.L.); and the Mainland-Hong Kong Joint Funding Scheme (MHP/030/20) (Y.L.); the Guangdong Province Engineering Laboratory for Biomedical Materials Additive Manufacturing (Y.L.).

## Author contributions

YC.Z., W.Z. and Y.L. designed the experiments, analyzed the data, and wrote the manuscript. YC.Z., A.G. synthesized and characterized the patches. C.L., A.G., Y.Y., Y.N., J.L., B.L., YM.Z., and L.L. helped with part of the material preparation and performed in vitro and in vivo experiments. YC.Z., W.Z., Z.C., and Y.L. discussed the results and edited the manuscript. Q.L., and Y.L. supervised the project. All authors discussed the results throughout the project and approved the final version of the manuscript.

## Competing interests

The authors declare no competing interests.
