## [Peer Review File · Nature Communications]

Reviewers' Comments:

Reviewer #1:

Remarks to the Author:

In this manuscript "Salting-in Effects of Black Phosphorus Nanosheets Enhance Water Absorption of Composite Patches for Robust Wet-tissue Adhesion", authors propose a novel composite patch with BP nanosheets (CPB) for wet-adhesion. The relative work is novelty and interesting.

Therefore, this manuscript should be considered for publication in Nature Communication, if the following questions can be addressed in the revision.

1. The composite patch with black phosphorus nanosheets (CPB) is the hydrogel or not?
2. In the in vivo phase, the group S+CPB+NIR were proved to prevent tumor recurrence in animal model, I doubt whether have a long-term mechanism is activated? Please explain it.
3. The scale bar in Figure 1j is missing.
4. More details about format should be checked.

Reviewer #2:

Remarks to the Author:

This work builds on previous work using black phosphorus nanosheets combined with non-covalently adhesive based on catechol (Biomater. Sci., 2023, 11, 235-247) to generate a biomaterial that is biodegradable, efficiently binds to wet tissue, and responds to near-infrared light to generate heat and reduce tumor recurrence. The chemical design is novel but has similarities to other published materials (Biomater Sci article above; also Small. 2022 Jul;18(26):e2201803). The work follows a model of describing wet adhesives, including as published in Nat Commun (<https://www.nature.com/articles/s41467-021-27529-5>), in (a) describing the synthesis and physicochemical/mechanical properties, and (b) showing successful proof-of-principle applications.

Strengths of the report include the demonstration that incorporating BP increases swelling by ~1.5-2x, increases tissue adhesion by ~0-3x, and the materials are well characterized in the supplement. The strength appears relatively high and a number of positive attributes are described compared to prior publications.

Given the similarity in concept to other published reports (<https://www.nature.com/articles/s41586-019-1710-5>, and others above), the impact of the manuscript may especially rely on the relative improvements of this approach compared to others. Gold-standard in vitro controls would be useful for the reader to compare in vitro material performance. Other wet-adhesive biomaterials have been developed and could be used.

The in vivo animal experiments are useful in showing a convincing effect of the materials, however the strength of the manuscript would be improved if the effects were compared to a gold-standard treatment, or potentially to show the impact of BP incorporation, since that is one of the key emphases in the novelty of the manuscript. Without such controls, it is difficult to assess the relative performance and any improvement in the application garnered from the design.

Figure 5 testing shows that NIR is required, but the control groups do not give evidence that the CPB material is required. For instance NIR-only control is lacking.

Is there any explanation why the adhesive effects of BP incorporation are different across different tissues? Especially pronounced for stomach but not heart.

Minor comments:

Suggest language editing service. First sentence "boomingly" is an example of odd phrasing.

Figures and figure captions may not meet guidelines for Nat Commun where statistical tests, individual-level data, p-values, and other metrics are reported.

Reviewer #3:

Remarks to the Author:

Zhang et al report a composite patch containing black phosphorus nanosheets (CPB). The authors show that CPB has higher bioadhesion than CP without BP nanosheets. They further demonstrate some bio-applications of the hydrogel patch for bleeding control, physical activating monitoring and tumor management. Despite showing certain levels of novelty, the reviewer believe that the current manuscript may be suitable for a more specialized journal and does not meet the high standard of Nature Communications. Some comments and suggestions to potentially improve the manuscript are listed below:

(1) Although the "salting in" effects and the resulted swelling, to a certain level, may contribute to the observed adhesion enhancement, the increased mechanical property of CPB compared to CP might be the main driving force. The reader wonder what would be the adhesion performance of both CPB and CP on dry tissues to decouple the effects of water (and salting in/swelling). As expected, PAA-DA seems to play a more prominent role regarding the adhesion increase, since both PAA and DA have been widely used in existing bioadhesives.

(2) For the bioapplications demonstrated, the advantage of the proposed CPB hydrogels compared to CP or other standard of care is not clear. In the in vivo hemostasis experiment, the authors only compared CPB with a negative control (without any treatment). It would be great if the authors could show whether this could be a good alternative or better option compared to existing treatment (e.g. hemostatic gauze or agent). For the physical activity monitoring, the authors claim that the CPB has better sensitivity due to the inclusion of BP nanosheets, but no evidence is provided. The demonstration of a combined therapy of CPB and NIR for tumor recurrence prevention is interesting, although it's unclear whether the enhanced bioadhesion would play a role.

(3) There're arbitrary claims in the manuscript that could be adjusted and typos to be corrected. In the Abstract, for example, "useless hydrogen bonds" could be replaced with better terms.

Response to Reviewers' Comments

Reviewer #1

General Comments:

In this manuscript “Salting-in Effects of Black Phosphorus Nanosheets Enhance Water Absorption of Composite Patches for Robust Wet-tissue Adhesion”, authors propose a novel composite patch with BP nanosheets (CPB) for wet-adhesion. The relative work is novelty and interesting. Therefore, this manuscript should be considered for publication in Nature Communication, if the following questions can be addressed in the revision.

Response:

Thanks so much for this reviewer for giving positive comment and insightful advice to help us improve the manuscript.

Comments:

Q1. The composite patch with black phosphorus nanosheets (CPB) is the hydrogel or not?

Response:

We thank for the reviewer's concern. CPB is the hydrogel which was first crosslinked to form polymer network as typical hydrogel and dried subsequently to form the patch, as shown in Fig. R1. With the presence of BP nanosheets, the enhanced water absorption capacity endowed CPB to achieve a robust wet-tissue adhesion.

Fig. R1. Schematic illustration of preparation of CPB. HAMA, PVA, BP, EDC, NHS and I2959 were mixed first. Gel solution was added into the mixture with fast stirring followed by being poured into a glass mold. The inter-crosslinking bonds between HAMA and Gel formed due to the presence of the EDC/NHS linker. Subsequently, the mixture was placed under the UV light, resulting in the self-crosslinking of HAMA. The hydroxyl groups of PVA could formed hydrogen bonds (HBs) with the functional groups such as NH- groups of HAMA and Gel to further enhance mechanical performance. The initial CPB product was obtained after the sample being dried overnight at room temperature (RT). The PAA-DA was dropwise added on the surface of the dried film to form the topological entanglement. Finally, CPB with triple crosslink network was prepared.

Q2. In the in vivo phase, the group S+CPB+NIR were proved to prevent tumor recurrence in animal model, I doubt whether have a long-term mechanism is activated? Please explain it.

Response:

We thank for the reviewer’s suggestion. The photothermal therapy (PTT) of CPB could kill remain tumour cells of the implant site at the initial stage and achieve a long-term inhibition for tumour recurrence. In Fig. R2, the *in vitro* studies showed that more than 95% tumour cells were apoptosis after NIR irradiation for 5 min in the CPB with NIR irradiation group compared to other groups (Refer to Supplementary Fig. 14, Page 18 of Supplementary Information). In addition, an *in vivo* study to estimate long-term tumour recurrence prevention of CPB with NIR

irradiation was carried out. As shown in Fig. R3, the S+CPB+NIR group had a long-term stable effect for preventing tumour recurrence for 4 weeks (Refer to Fig. 5 and Page 16 of Revision).

Fig. R2. *In vitro* cell studies. (a) Fluorescence images of tumour cells after adding the patches without and with NIR irradiation (808 nm, $P=1 \text{ W cm}^{-2}$) for 5 min. The live cells are in green and dead cells are in red. (b) Representative Annexin V-FITC/PI scatter plots of MCF-7 cells after 6 h of treatment. (c) Cell proliferation of L929 cells. Values represent the mean and standard deviation, $n = 4$ independent samples.

Fig. R3. *In vivo* studies of tumour postsurgical treatment. **a** Treatment schedule of tumour inoculation, resection and patch implantation. **b** Thermal images (left) and temperature changes (right) at the surgical site of tumour-bearing mice with CPB under NIR light (808 nm, 1 W cm⁻²). Values represent the mean and standard deviation, $n = 3$ independent samples. **c** Body weight changes of the nude mice recorded every two days in various groups. S: only surgical resection; NIR: after surgical resection, the surgical site was exposed to NIR light (808 nm, 1 W cm⁻²) for 5 minutes; S+CP: after surgical resection, CP was adhered to the surgical site; S+CPB: after surgical resection, CPB was adhered to the surgical site without NIR light; S+CPB+NIR: after surgical resection, CPB was adhered to the surgical site subsequently with NIR light (808 nm, 1 W cm⁻²). Values represent the mean and standard deviation, $n = 3$ independent samples. **d** Tumor volume after various treatments recorded every two days in various groups. Values represent the mean and standard deviation, $n = 3$ independent samples. **e** Fluorescence images of the tumour-bearing mice with various treatments immediately before and after surgery, and at 1 week, 2 weeks, 3 weeks, and 4 weeks. **f** H&E-stained sections of major organs from the sacrificed nude mice in each group after 4 weeks.

Q3. The scale bar in Figure 1j is missing.

Response:

We thank for the reviewer's suggestion. After carefully checked, we found the scale bar in Fig. 2J was missing. We magnified the scale bars and inserted in Fig. 2J as shown in Fig. R4 (Refer to Fig. 2, Page 8 of Revision).

Fig. R4. Structures of the patches and the swelling capacity of CPB. **a** Images of patches with various contents of BP nanosheets: CP: 0 mg, CPB-0.2: 0.2 mg, CPB-0.6: 0.6 mg, CPB: 1 mg, CPB-1.2: 1.2 mg; flexible adhesion and easy storage of the patches. **b** Schematic illustration of the degradation of the BP nanosheets and proposed structures of CP and CPB. **c** Binding energies of nitrogen (N1s) in the XPS spectrum of CPB. **d** Binding energies of phosphorus (P2p) in the XPS spectrum of CPB. **e** Swelling capacity of CP and CPB in 60 minutes at RT. Values represent the mean and standard deviation, $n = 3$ independent samples. **f** Water vapor sorption capacity of CP and CPB in 60 minutes in an environmental chamber with 90-95% RH at RT. Values represent the mean and standard deviation, $n = 3$ independent samples. **g** Contact angles

of water drops on the patches in 10 minutes. **h** Confocal fluorescence microscopy images of the patches before and after being swelled for 1 minute. Green, mixture of fluorescent dye (calcein). “Cloud”, the fluid being swelled. **i** Infrared thermal images of the patches in hot steam. **j** SEM images (60 x and 500 x) of the patches after being lyophilized.

Q4. More details about format should be checked.

Response:

We thank for the reviewer’s suggestion. The format has been checked throughout the manuscript and the revisions are highlighted, as shown below:

- (a) The title of this manuscript was revised to 10 words and highlighted in revised manuscript:
Composite Patches Integrated with Black Phosphorus for Robust Wet-tissue Adhesion.
- (b) The format of the figures and figure captions was corrected.
- (c) The format of the subheadings was corrected.

Reviewer #2

Comments:

Q1. This work builds on previous work using black phosphorus nanosheets combined with non-covalently adhesive based on catechol (Biomater. Sci., 2023, 11, 235-247) to generate a biomaterial that is biodegradable, efficiently binds to wet tissue, and responds to near-infrared light to generate heat and reduce tumor recurrence. The chemical design is novel but has similarities to other published materials (Biomater Sci article above; also Small. 2022 Jul;18(26):e2201803). The work follows a model of describing wet adhesives, including as published in Nat Commun (<https://www.nature.com/articles/s41467-021-27529-5>), in (a) describing the synthesis and physicochemical/mechanical properties, and (b) showing successful proof-of-principle applications.

Response:

We really appreciate for the reviewer's valuable comment. The mentioned work (*Biomater. Sci., 2023, 11, 235-247*) developed a hydrogel adhesive patch containing the black phosphorus nanosheets (BPNSs) for effective infectious burn wound healing. The other work (*Small. 2022 Jul;18(26):e2201803*) took advantages of the drug loading ability and photothermal effects of the BPNSs to design a nanoplatform with enhanced antitumour abilities. Although these papers have proposed BPNSs-contained biomaterials with good performances, our CPB is different in material design, adhesive mechanism and application, specifically as follows.

Different material design strategy. The design of the adhesive patch in *Biomater. Sci. 2023* was on the basis of the Schiff base reaction between the amino residues on CA-CS and the aldehyde groups on Odex. The fabricated BP-DOX@Gd/(DOPA)₄-PEG-TL nanoplatform in *Small. 2022* was mainly based on the various surface modification strategies of the BPNS or the peptide mimics. In our manuscript, the proposed CPB with triple crosslink network was designed and fabricated based on inter-crosslinking between HAMA and Gel, the self-crosslinking of HAMA, as well as the surface modification using PAA-DA for further improving the mechanical and adhesive properties.

Different adhesive mechanism and applications. The wet adhesion of the proposed patch in *Biomater. Sci., 2023* was attributed to the catechol modification and the crosslinker Odex, which was used as a bio-sealant for subsequent epithelial-to-mesenchymal transition (EMT). The adhesion of biomimetic peptide in *Small. 2022* was used for surface modification during synthesis, but not for adhering to wet tissues; then the proposed nanoplatform employed

photothermal effects of the BPNSs for enhancing synergetic antitumour abilities. In our manuscript, the robust adhesion of CBP was attributed to the enhanced water-absorption capacity originating from BP. The BP nanosheets also endowed CPB with robust mechanical properties, good electrical conductivity and photothermal effect to achieve multiple bioapplications including rapid hemostasis, physical-activity monitoring and effective tumour-recurrence prevention.

For the work published in *Nat Commun* (<https://www.nature.com/articles/s41467-021-27529-5>), the relevant description and methodology were good references for the projects related to wet adhesives. The similar expressions of the synthesis, property characterization, and applications were also employed in some other published work e.g., *Nature* 575, 169–174 (2019), *Sci. Transl. Med.* 14, eabh2857(2022) and *Nature Mater* 15, 190–196 (2016). In addition, our manuscript focused on the effects of the enhanced water-absorption capacity of CPB, so we presented a series of experimental results related to structures and water absorption capacity of the patches. We have also cited the prior work the reviewer recommended and highlighted them in revised manuscript. (Refer to Reference No. 15, 23, 24, and the section of Introduction of Revision).

Q2. Strengths of the report include the demonstration that incorporating BP increases swelling by ~1.5-2x, increases tissue adhesion by ~0-3x, and the materials are well characterized in the supplement. The strength appears relatively high and a number of positive attributes are described compared to prior publications.

Response:

We deeply appreciate for the reviewer's positive comments.

Q3. Given the similarity in concept to other published reports (<https://www.nature.com/articles/s41586-019-1710-5>, and others above), the impact of the manuscript may especially rely on the relative improvements of this approach compared to others. Gold-standard in vitro controls would be useful for the reader to compare in vitro material performance. Other wet-adhesive biomaterials have been developed and could be used.

Response:

We really appreciate for the reviewer's valuable comment. For the studies on wet-tissue adhesives, the cited report and other pervious work usually focus on enhancing the mechanical properties of adhesives and improving the molecular interaction between adhesives and tissues. However, the substantially different concept was raised in this manuscript, of which the improved adhesive performances were based on enhancing the water-absorption capacity of the patches. As proved, we found that CPB with enhanced water absorption capacity finally achieved robust wet-adhesion. We also used commercial cyanoacrylate tissue adhesives as the control groups to evaluate adhesive properties (Fig. R5, Refer to Supplementary Fig. 12, Page 15 of Supplementary Information) and added the results in our revised manuscript.

Revision:

[Page 10, Paragraph 1]

“For further comparison, we used commercial cyanoacrylate tissue adhesives as control groups to evaluate the adhesive effects (Supplementary Fig. 12). As presented, the shear stress and interfacial toughness of commercial adhesive 1 were ~ 74 KPa and ~ 73 N/m for wet porcine skin, while those of commercial adhesive 2 were ~ 105 KPa and ~ 88 N/m for wet porcine skin. CPB presented significantly enhanced adhesion to various wet tissues compared with the commercial adhesives...”

Fig. R5. Adhesive performance of CPB and the commercial tissue adhesives adhered to wet porcine tissues from various organs (skin, heart, stomach, liver). (a) Shear stress. (b) Interfacial toughness. Values represent the mean and standard deviation, $n = 3$ independent samples. *, significant difference compared to the commercial tissue adhesives, $p < 0.05$.

Q4. The *in vivo* animal experiments are useful in showing a convincing effect of the materials, however the strength of the manuscript would be improved if the effects were compared to a gold-standard treatment, or potentially to show the impact of BP incorporation, since that is one of the key emphases in the novelty of the manuscript. Without such controls, it is difficult to assess the relative performance and any improvement in the application garnered from the design.

Response:

We thank for the reviewer’s valuable suggestion. As shown in Fig. R6 (Refer to Supplementary Fig. 15, Page 20 of Supplementary Information) and Fig. R7 (Refer to Fig. 4 and Page 14 of Revision), we supplemented comparative experiments on hemostasis evaluation *in vivo*, using commercial products including the gauze or gelatin sponge as the control group; and the physical activity monitoring studies using the CP, CPB and Cu foil as EMG sensors.

Fig. R6. *In vivo* hemostasis. (a) Images of the hemostatic effect for the damaged liver in the blank group, CPB group, CP group, Gelatin Sponge group and Gauze group. Yellow dotted line: the position of the wound or the sample. (b) Bloodstain on the surface of the filter paper in the blank group, CPB group, CP group, Gelatin Sponge group and Gauze group at 120 seconds. (c) Total blood loss in the blank group, CPB group, CP group, Gelatin Sponge group and Gauze group. Values represent the mean and standard deviation, $n = 3$ independent samples. *, significant difference compared to the blank group, Gelatin Sponge group and Gauze group, $p < 0.05$. (d) Images of the hemostatic effect of the CP, Gelatin Sponge and Gauze in a rat dynamic heart perforation wound model. Yellow dotted line: the position of the wound or the sample.

Fig. R7. Potential applications in *in vivo* hemostasis and activity monitoring. a Schematic illustration of the hemostatic process using CPB in a rat liver perforation wound model. b

Images of the hemostatic effect on damaged livers in the blank group and the CPB group. Yellow dotted line: position of the wound or CPB. **c** Bloodstain on the surface of filter paper in the blank group and the CPB group at 120 seconds. **d** Total blood loss in the blank group and the CPB group. Values represent the mean and standard deviation, $n=3$ independent samples. *, significant difference compared to the blank group, $p<0.05$. **e** Images of the hemostatic effect of CPB in a rat dynamic heart perforation wound model. Yellow dotted line: the position of the wound or the CPB. **f** Schematic illustration of CPB as a physical activity monitor and the adhesion of the CPB monitor on a nude mouse. **g** Potentials of the nude mouse from under anesthetic to wake up state.

The description and results of various control groups for *in vivo* studies assays have also been added and highlighted in revised manuscript and Supplementary Information.

Revision:

[Page 12, Paragraph 1]

“First, we used a normal SD rat liver perforation wound model to investigate the hemostatic effect of CPB (Fig. 4a and Supplementary Movie 3). CP and commercial products (e.g., Gelatin Sponge and Gauze) were used as the control groups (Supplementary Fig. 15). After perforation, the sample was directly adhered to the wound, while no treatment was applied in the blank group. For the CPB group, only a small area of bloodstain appeared on the surface of the filter paper beneath the liver upon patch adhesion to the wound (approximately 1-2 seconds) (Fig. 4b). For the CP and Gelatin Sponge groups, the area of bloodstain was slightly greater compared to that in the CPB group. In contrast, a clear bleeding pathway was observed in the Gauze group and the blank group (Supplementary Fig. 15a, Fig. 4b). Over time, the bloodstains in the Gauze group and blank group became more obvious, but there was minimal change in the CPB group (Supplementary Fig. 15a and b, Fig. 4b and c). Quantitatively, the total blood loss of the CPB group (~0.07 g) was much less than the blank group (~0.4 g) and the commercial product groups (~0.11-0.13 g) (Supplementary Fig. 15c, Fig. 4d). Additionally, a dynamic heart perforation wound model was used, where CPB was rapidly adhered to the wound and then pressed for 3-5 seconds after perforation (Fig. 4e). The hemostatic effect of CP and the commercial products in the control groups can be found in Supplementary Fig. 15d. Once the sample was positioned, no further obvious blood was observed around the wound in the Gelatin Sponge group, the CP group and the CPB group, but it did in the Gauze group. Even acting on a dynamic and curved

surface, CPB still exhibited a swift hemostatic effect, which can be attributed to its strong adhesion ability reducing blood diffusion and strengthening the wound seal.”

[Page 13, Paragraph 2]

“With its conductive properties, CPB can be directly adhered to the skin of a nude mouse and used as a physical activity monitor (Fig. 4f). Fig. 4g illustrates the electromyographic (EMG) signals acquired by CP, CPB and Cu foil of the nude mouse from under anesthetic to wake up state. CPB and Cu foil presented the similar level of sensing capability, which was superior to that of the CP sample. The physiological activities of wake-up processes could be clearly identified through the amplitude and frequency of the sensing potentials. The presence of BP nanosheets improved the electrical conductivity and sensitivity of the patches, allowing CPB to potentially be popularized as a biomedical sensor.”

Supplementary Information:

[Page 19, Paragraph 2; Supplementary Fig. 15]

Potential application of hemostasis and activity monitor

“CP and the commercial products (e.g. Gelatin Sponge and Gauze) were used as control groups for evaluating the hemostatic effect in a normal SD rat liver perforation wound model. A total of 15 SD rats (male, weight of 250-300 g, 7-8 weeks) were randomly divided into the CP group, Gelatin Sponge group, and Gauze group. Then the livers of the rats were lifted and placed on the surface of preweighted filter paper, and a circular perforation wound (diameter of 6 mm) was created for hemorrhage. The sample was cut to a size of $10 \times 10 \times 0.2$ mm and weighted in advance. Next the corresponding sample in each group was directly adhered to the bleeding site and the hemostatic process was recorded with a digital camera. The blood loss was calculated by determining the total weight of the blood absorbed by the filter paper and the sample, respectively. Similarly, the hemostatic effect of CP, the commercial Gelatin Sponge and Gauze were evaluated by a normal SD rat heart perforation wound model, where the hearts of rats were lifted and a circular perforation wound (diameter of 6 mm) was created for hemorrhage. The corresponding sample ($10 \times 10 \times 0.2$ mm) in each group was immediately adhered to the bleeding sites, and the state was recorded with a digital camera. (Supplementary Fig. 15).”

Q5. Figure 5 testing shows that NIR is required, but the control groups do not give evidence that the CPB material is required. For instance NIR-only control is lacking.

Response:

We really appreciate the reviewer for pointing out this. As shown in Fig. R8 (Refer to Fig. 5 and Page 16 of Revision), the NIR-only control group indicated that NIR light alone did not inhibit tumour growth. We have also included the relevant results in revised manuscript.

Revision:

[Page 15, Paragraph 2]

“Tumour recurrence in the NIR group was similar to that in the S group, indicating that NIR light alone did not inhibit tumour growth...”

[Page 20, Paragraph 4].

“A total of 36 mice were randomly divided into six groups for investigating histocompatibility and tumour recurrence prevention: blank group (no surgical treatment), surgery group (S group: only surgical removal of the tumour, no materials), surgery with NIR light only (NIR: after surgical resection, the surgical site was exposed to NIR light (808 nm, 1 W cm⁻²) for 5 minutes), surgery with the CP group...”

Fig. R8. *In vivo* studies of tumour postsurgical treatment. **a** Treatment schedule of tumour inoculation, resection and patch implantation. **b** Thermal images (left) and temperature changes (right) at the surgical site of tumour-bearing mice with CPB under NIR light (808 nm, 1 W cm⁻²). Values represent the mean and standard deviation, $n = 3$ independent samples. **c** Body weight changes of the nude mice recorded every two days in various groups. S: only surgical resection; NIR: after surgical resection, the surgical site was exposed to NIR light (808 nm, 1 W cm⁻²) for 5 minutes; S+CP: after surgical resection, CP was adhered to the surgical site; S+CPB: after surgical resection, CPB was adhered to the surgical site without NIR light; S+CPB+NIR: after surgical resection, CPB was adhered to the surgical site subsequently with NIR light (808 nm, 1 W cm⁻²). Values represent the mean and standard deviation, $n = 3$ independent samples. **d** Tumour volume after various treatments recorded every two days in various groups. Values represent the mean and standard deviation, $n = 3$ independent samples. **e** Fluorescence images of the tumour-bearing mice with various treatments immediately before and after surgery, and at 1 week, 2 weeks, 3 weeks, and 4 weeks. **f** H&E-stained sections of major organs from the sacrificed nude mice in each group after 4 weeks.

Q6. Is there any explanation why the adhesive effects of BP incorporation are different across different tissues? Especially pronounced for stomach but not heart.

Response:

We thank for the review's comment. In terms of different tissues, the adhesive effect depends on the surface condition, including surface texture, category and the amount of surface liquid, functional groups in the tissue proteins (*Prog. Polym. Sci.*, 39.7 (2014): 1375-1405). For the tissue itself, also the mechanical properties like dynamic behaviors, flexibility, elasticity and modulus can generate large impacts (*Nat. Rev. Mater.*, 5.4 (2020): 310-329). Thus, the adhesive effects of BP incorporation could be synergistically affected by various factors. As shown in Fig. R9 (Refer to Fig. 3 and Page 11 of Revision), the significant improvement due to BP incorporation appeared in stomach-adhesion group rather than the heart-adhesion group using lap-shear measurement, but the same trend appeared in both stomach-adhesion group and heart-adhesion group using the modified 180° peel measurement. In this study, we mainly focused on the comprehensive influence of the BP incorporation on wet adhesion and the robust adhesion of CPB for different biomedical apparatuses. The results have demonstrated that introducing BP nanosheets could enhance the adhesive properties of CPB with wet tissues. For further comparison, we have supplemented the commercial cyanoacrylate tissue adhesives as control groups to evaluate the adhesive effects (Fig. R10, Refer to Supplementary Fig. 12, Page 15 of Supplementary Information). Even the test data of the heart adhesion was lower than the stomach adhesion using CPB, the shear stress (~44 KPa) of CPB adhered to heart were higher than those of commercial products (~35 KPa and ~21 KPa, respectively). In addition, the hemostatic effect studies *in vivo* proved CPB can adhered stably on the dynamic surface of the heart to achieve a good hemostatic effect (Fig. R11, Refer to Fig. 4 and Page 14 of Revision).

Fig. R9. **a** Schematic representation (left) of the lap-shear measurement process and photograph (right) of CPB adhered to tissue for this measurement. **b** Shear stress of CP and CPB (with or without PAA-DA) adhered to porcine tissues from various organs (skin, heart, stomach, liver). Values represent the mean and standard deviation, $n = 3$ independent samples. *, significant difference compared to the CP group, $p < 0.05$. **c** Schematic representation (left) of the modified 180° peel measurement process and photograph (right) of CPB adhered to tissue for this measurement. **d** Interfacial toughness of CP and CPB (with or without PAA-DA) adhered to

porcine tissues from various organs (skin, heart, stomach, liver). Values represent the mean and standard deviation, $n = 3$ independent samples. *, significant difference compared to the CP group, $p < 0.05$. **e** Lap-shear (top) and modified 180° peel (bottom) adhesive performances of CP and CPB (with PAA-DA) adhered to skin tissues from nude mice. Values represent the mean and standard deviation, $n = 3$ independent samples. *, significant difference compared to the CP group, $p < 0.05$. **f** Schematic illustration of the enhanced adhesion mechanism of CPB to wet tissues. **g** Effect of adhesion time on the adhesive performance of the patches. Values represent the mean and standard deviation, $n = 3$ independent samples. *, significant difference compared to the CP group, $p < 0.05$. **h and i** Images of CPB adhered to a series of representative tissues (**h**) and to bone tissue coated with blood (**i**). Red arrows: CPB. **j** Sealing of a fluid-leaking *ex vivo* rabbit stomach by CPB.

Fig. R10. Supplementary Fig. 12 Adhesive performance of CPB and the commercial tissue adhesives adhered to wet porcine tissues from various organs (skin, heart, stomach, liver). (a) Shear stress. (b) Interfacial toughness. Values represent the mean and standard deviation, $n = 3$ independent samples. *, significant difference compared to the commercial tissue adhesives, $p < 0.05$.

Fig. R11. Potential applications in *in vivo* hemostasis and activity monitoring. **a** Schematic illustration of the hemostatic process using CPB in a rat liver perforation wound model. **b** Images of the hemostatic effect on damaged livers in the blank group and the CPB group. Yellow dotted line: position of the wound or CPB. **c** Bloodstain on the surface of filter paper in the blank group and the CPB group at 120 seconds. **d** Total blood loss in the blank group and the CPB group. Values represent the mean and standard deviation, $n=3$ independent samples. *, significant difference compared to the blank group, $p<0.05$. **e** Images of the hemostatic effect of CPB in a rat dynamic heart perforation wound model. Yellow dotted line: the position of the wound or the CPB. **f** Schematic illustration of CPB as a physical activity monitor and the adhesion of the CPB monitor on a nude mouse. **g** Potentials of the nude mouse from under anesthetic to wake up state.

Minor comments:

Q7. Suggest language editing service. First sentence “boomingly” is an example of odd phrasing.

Response:

We really appreciate the reviewer for pointing out this. The word has been revised and highlighted in revised manuscript:

Revision:

[Page 3, Paragraph 1]

“Tissue adhesives have been extensively developed in biomedical fields due to their ability to halt bleeding...”.

The revised manuscript was polished for proper English language, grammar, etc. by Springer Nature Author Services with a certification (verification code 1C8D-C677-E47E-4C6E-BAEP).

Q8. Figures and figure captions may not meet guidelines for Nat Commun where statistical tests, individual-level data, p-values, and other metrics are reported.

Response:

We really appreciate the reviewer for pointing out this. The format of the revised manuscript has been supplemented and highlighted as shown below:

- (a) Fig. 2: “...**e** Swelling capacity of CP and CPB in 60 minutes at RT. Values represent the mean and standard deviation, $n = 3$ independent samples. **f** Water vapor sorption capacity of CP and CPB in 60 minutes in an environmental chamber with 90-95% RH at RT. Values represent the mean and standard deviation, $n = 3$ independent samples....”.
- (b) Fig. 3: “...**b** Shear stress of CP and CPB (with or without PAA-DA) adhered to porcine tissues from various organs (skin, heart, stomach, liver). Values represent the mean and standard deviation, $n = 3$ independent samples. *, significant difference compared to the CP group, $p < 0.05$. **c** Schematic representation (left) of the modified 180° peel measurement process and photograph (right) of CPB adhered to tissue for this measurement. **d** Interfacial toughness of CP and CPB (with or without PAA-DA) adhered to porcine tissues from various organs (skin, heart, stomach, liver). Values represent the mean and standard deviation, $n = 3$ independent samples. *, significant difference compared to the CP group,

$p < 0.05$. **e** Lap-shear (top) and modified 180° peel (bottom) adhesive performances of CP and CPB (with PAA-DA) adhered to skin tissues from nude mice. Values represent the mean and standard deviation, $n = 3$ independent samples. *, significant difference compared to the CP group, $p < 0.05$. **f** Schematic illustration of the enhanced adhesion mechanism of CPB to wet tissues. **g** Effect of adhesion time on the adhesive performance of the patches. Values represent the mean and standard deviation, $n = 3$ independent samples. *, significant difference compared to the CP group, $p < 0.05$...”

(c) Fig. 4: “...**d** Total blood loss in the blank group and the CPB group. Values represent the mean and standard deviation, $n = 3$ independent samples. *, significant difference compared to the blank group, $p < 0.05$...”

(d) Fig. 5: “...**b** Thermal images (left) and temperature changes (right) at the surgical site of tumour-bearing mice with CPB under NIR light (808 nm , 1 W cm^{-2}). Values represent the mean and standard deviation, $n = 3$ independent samples. **c** Body weight changes of the nude mice recorded every two days in various groups. S: only surgical resection; NIR: after surgical resection, the surgical site was exposed to NIR light (808 nm , 1 W cm^{-2}) for 5 minutes; S+CP: after surgical resection, CP was adhered to the surgical site; S+CPB: after surgical resection, CPB was adhered to the surgical site without NIR light; S+CPB+NIR: after surgical resection, CPB was adhered to the surgical site subsequently with NIR light (808 nm , 1 W cm^{-2}). Values represent the mean and standard deviation, $n = 3$ independent samples. **d** Tumour volume after various treatments recorded every two days in various groups. Values represent the mean and standard deviation, $n = 3$ independent samples...”

(e) Supplementary Fig. 7: “... **(b)** Young’s modulus, tensile strength and elongation at break derived from (a). Values represent the mean and standard deviation, $n = 3$ independent samples. *, significant difference compared to the CP group, $p < 0.05$. **(c)** Stress-stain curves of CPB with various ratios of PVA (left), various graft-ratios of HAMA (middle), various ratios of BP nanosheets (right), respectively. **(d)** Mechanical data derived from (c), respectively. Values represent the mean and standard deviation, $n = 3$ independent samples...”

(f) Supplementary Fig. 8: “Electrical resistivity of CP and CPB with various contents of BP nanosheets: CP: 0 mg, CPB-0.2: 0.2 mg, CPB-0.6: 0.6 mg, CPB: 1 mg, CPB-1.2: 1.2 mg. Values represent the mean and standard deviation, $n = 3$ independent samples.”

(g) Supplementary Fig. 9: “Degradation *in vitro* of CP and CPB in 60 days. Values represent the mean and standard deviation, $n = 3$ independent samples.”

(h) Supplementary Fig. 10: “**(a)** Photothermal heating curves of CP and CPB at dry state and

wet state irradiated by the NIR laser (808 nm, 1 W cm⁻²). Insert: infrared thermal images of CPB at 60 seconds. Values represent the mean and standard deviation, $n = 3$ independent samples...”

- (i) Supplementary Fig. 14: “...**(c)** Cell proliferation of L929 cells. Values represent the mean and standard deviation, $n = 4$ independent samples.”
- (j) Supplementary Fig. 17: “... **(b)** Body weight changes of the nude mice recorded every two days. Values represent the mean and standard deviation, $n = 3$ independent samples. **(c)** Tumour volume changes recorded every two days. Values represent the mean and standard deviation, $n = 3$ independent samples.”
- (k) Supplementary Fig. 18: “...CP group: implanting CP; CPB group: implanting CPB; CPB+NIR group: implanting CPB and irradiated by NIR light (808 nm, P=1 W cm⁻²) for 5 minutes. Values represent the mean and standard deviation, $n = 3$ independent samples.”

Reviewer #3

General Comments:

Zhang et al report a composite patch containing black phosphorus nanosheets (CPB). The authors show that CPB has higher bioadhesion than CP without BP nanosheets. They further demonstrate some bio-applications of the hydrogel patch for bleeding control, physical activating monitoring and tumor management. Despite showing certain levels of novelty, the reviewer believe that the current manuscript may be suitable for a more specialized journal and does not meet the high standard of Nature Communications. Some comments and suggestions to potentially improve the manuscript are listed below:

Response:

Thanks so much for this reviewer's valuable comment. As suggested, we supplemented **(1)** further adhesion tests *in vitro* and **(2)** hemostasis evaluation *in vivo* using the commercial products, including the cyanoacrylate tissue adhesives, the gauze or gelatin sponge, as the control groups; as well as **(3)** the physical activity monitoring comparison studies using the CP, CPB and Cu foil. **(4)** We removed the description about salting-in effect but focused on the enhanced water absorption capacity of CPB for clearly presenting adhesive mechanism. **(5)** We have also addressed all the specific comments raised by this reviewer.

Q1. Although the "salting in" effects and the resulted swelling, to a certain level, may contribute to the observed adhesion enhancement, the increased mechanical property of CPB compared to CP might be the main driving force. The reader wonder what would be the adhesion performance of both CPB and CP on dry tissues to decouple the effects of water (and salting in/swelling). As expected, PAA-DA seems to play a more prominent role regarding the adhesion increase, since both PAA and DA have been widely used in existing bioadhesives.

Response:

We thank the reviewer's comment. We have evaluated the effects of the PAA-DA on the adhesive performances in this study. The shear stress and interfacial toughness of CPB without PAA-DA adhered to wet porcine skin were ~119 KPa and ~422 N/m, while those of CP without PAA-DA were ~66 KPa and ~378 N/m. Even PAA-DA could further improve the adhesive performance of the patches, CPB has already presented good adhesive performance to wet tissues without adding PAA-DA (Fig. R12, Refer to Fig. 3 and Page 11 of Revision). In this

situation, CPB had a large enhancement compared with CP in wet adhesion as well.

As suggested, the adhesive performances of both CPB and CP on dry tissues have been evaluated. As shown in Fig. R13 (Refer to Supplementary Fig. 13, Page 16 of Supplementary Information), the enhanced water absorption capacity played more important role in wet adhesion of CPB than the increased mechanical property of CPB. The results and description have been incorporated and highlighted in our revised supplementary information.

Supplementary Information:

[Page 16, Paragraph 1]

“In addition, the adhesive performance of the patches adhered to dry porcine skin after 12 h were also evaluated. The shear stress of CP and CPB in the dry tissues group was ~1 KPa, which had no significant difference (Supplementary Fig. 13a). In the wet tissues group, the shear stress of CPB (~80 KPa) was significantly higher compared with CP (~60 KPa) (Supplementary Fig. 13b). The stress difference between the dry and wet tissues groups should be attributed to water content of the patches⁵. The experimental results indicated that the enhanced water absorption capacity contributed more in wet adhesion of CPB than the increased mechanical property of CPB.”

Fig. R12. Adhesive performances. **a** Schematic representation (left) of the lap-shear measurement process and photograph (right) of CPB adhered to tissue for this measurement. **b** Shear stress of CP and CPB (with or without PAA-DA) adhered to porcine tissues from various organs (skin, heart, stomach, liver). Values represent the mean and standard deviation, $n = 3$ independent samples. *, significant difference compared to the CP group, $p < 0.05$. **c** Schematic representation (left) of the modified 180° peel measurement process and photograph (right) of CPB adhered to tissue for this measurement. **d** Interfacial toughness of CP and CPB (with or

without PAA-DA) adhered to porcine tissues from various organs (skin, heart, stomach, liver). Values represent the mean and standard deviation, $n = 3$ independent samples. *, significant difference compared to the CP group, $p < 0.05$. **e** Lap-shear (top) and modified 180° peel (bottom) adhesive performances of CP and CPB (with PAA-DA) adhered to skin tissues from nude mice. Values represent the mean and standard deviation, $n = 3$ independent samples. *, significant difference compared to the CP group, $p < 0.05$. **f** Schematic illustration of the enhanced adhesion mechanism of CPB to wet tissues. **g** Effect of adhesion time on the adhesive performance of the patches. Values represent the mean and standard deviation, $n = 3$ independent samples. *, significant difference compared to the CP group, $p < 0.05$. **h and i** Images of CPB adhered to a series of representative tissues (**h**) and to bone tissue coated with blood (**i**). Red arrows: CPB. **j** Sealing of a fluid-leaking *ex vivo* rabbit stomach by CPB.

Fig. R13. Adhesive performance of CP and CPB to dry porcine skin (a) and wet porcine skin (b) after 12 h. Values represent the mean and standard deviation, $n = 3$ independent samples. *, significant difference compared to the CP group, $p < 0.05$.

*Q2. For the bioapplications demonstrated, the advantage of the proposed CPB hydrogels compared to CP or other standard of care is not clear. In the *in vivo* hemostasis experiment, the authors only compared CPB with a negative control (without any treatment). It would be great if the authors could show whether this could be a good alternative or better option compared to existing treatment (e.g. hemostatic gauze or agent). For the physical activity monitoring, the authors claim that the CPB has better sensitivity due to the inclusion of BP nanosheets, but no evidence is provided. The demonstration of a combined therapy of CPB and NIR for tumor recurrence prevention is interesting, although it's unclear whether the enhanced bioadhesion*

would play a role.

Response:

We appreciate the reviewer's helpful suggestion. The detailed responses and revisions are separately presented below.

(1) For *in vivo* hemostasis experiment. The experiments of various control groups for *in vivo* studies assays have been supplemented (Fig. R14, Refer to Supplementary Fig. 15, Page 20 of Supplementary Information)) and the related results were added and highlighted in our revised manuscript and Supplementary Information.

Revision:

[Page 12, Paragraph 1]

“First, we used a normal SD rat liver perforation wound model to investigate the hemostatic effect of CPB (Fig. 4a and Supplementary Movie 3). CP and commercial products (e.g., Gelatin Sponge and Gauze) were used as the control groups (Supplementary Fig. 15). After perforation, the sample was directly adhered to the wound, while no treatment was applied in the blank group. For the CPB group, only a small area of bloodstain appeared on the surface of the filter paper beneath the liver upon patch adhesion to the wound (approximately 1-2 seconds) (Fig. 4b). For the CP and Gelatin Sponge groups, the area of bloodstain was slightly greater compared to that in the CPB group. In contrast, a clear bleeding pathway was observed in the Gauze group and the blank group (Supplementary Fig. 15a, Fig. 4b). Over time, the bloodstains in the Gauze group and blank group became more obvious, but there was minimal change in the CPB group (Supplementary Fig. 15a and b, Fig. 4b and c). Quantitatively, the total blood loss of the CPB group (~0.07 g) was much less than the blank group (~0.4 g) and the commercial product groups (~0.11-0.13 g) (Supplementary Fig. 15c, Fig. 4d). Additionally, a dynamic heart perforation wound model was used, where CPB was rapidly adhered to the wound and then pressed for 3-5 seconds after perforation (Fig. 4e). The hemostatic effect of CP and the commercial products in the control groups can be found in Supplementary Fig. 15d. Once the sample was positioned, no further obvious blood was observed around the wound in the Gelatin Sponge group, the CP group and the CPB group, but it did in the Gauze group. Even acting on a dynamic and curved surface, CPB still exhibited a swift hemostatic effect, which can be attributed to its strong adhesion ability reducing blood diffusion and strengthening the wound seal.”

Supplementary Information:

[Page 19, Paragraph 2; Supplementary Fig. 15]

Potential application of hemostasis and activity monitor

“CP and the commercial products (e.g. Gelatin Sponge and Gauze) were used as control groups for evaluating the hemostatic effect in a normal SD rat liver perforation wound model. A total of 15 SD rats (male, weight of 250-300 g, 7-8 weeks) were randomly divided into the CP group, Gelatin Sponge group, and Gauze group. Then the livers of the rats were lifted and placed on the surface of preweighted filter paper, and a circular perforation wound (diameter of 6 mm) was created for hemorrhage. The sample was cut to a size of $10 \times 10 \times 0.2$ mm and weighted in advance. Next the corresponding sample in each group was directly adhered to the bleeding site and the hemostatic process was recorded with a digital camera. The blood loss was calculated by determining the total weight of the blood absorbed by the filter paper and the sample, respectively. Similarly, the hemostatic effect of CP, the commercial Gelatin Sponge and Gauze were evaluated by a normal SD rat heart perforation wound model, where the hearts of rats were lifted and a circular perforation wound (diameter of 6 mm) was created for hemorrhage. The corresponding sample ($10 \times 10 \times 0.2$ mm) in each group was immediately adhered to the bleeding sites, and the state was recorded with a digital camera. (Supplementary Fig. 15).”

Fig. R14. *In vivo* hemostasis. (a) Images of the hemostatic effect for the damaged liver in the blank group, CPB group, CP group, Gelatin Sponge group and Gauze group. Yellow dotted line: the position of the wound or the sample. (b) Bloodstain on the surface of the filter paper in the blank group, CPB group, CP group, Gelatin Sponge group and Gauze group at 120 seconds. (c) Total blood loss in the blank group, CPB group, CP group, Gelatin Sponge group and Gauze group. Values represent the mean and standard deviation, $n = 3$ independent samples. *, significant difference compared to the blank group, Gelatin Sponge group and Gauze group, $p < 0.05$. (d) Images of the hemostatic effect of the CP, Gelatin Sponge and Gauze in a rat dynamic heart perforation wound model. Yellow dotted line: the position of the wound or the sample.

(2) For the physical activity monitoring. The electrical resistivity of CPB has been compared to CP and Cu. As shown in Fig. R15 (Refer to Supplementary Fig. 8, Page 11 of Supplementary Information), electrical resistivity of CPB noticeably decreased by incorporating BP nanosheets,

and even reached the similar value of Cu foil. In addition, we also supplemented the physical activity monitoring studies on the nude mouse using the CP, CPB and Cu foil as EMG sensors (Fig. R16, Refer to Fig. 4 and Page 14 of Revision). We have included the results and relevant discussion in our revised manuscript.

Fig. R15. Electrical resistivity of CP and CPB with various contents of BP nanosheets: CP: 0 mg, CPB-0.2: 0.2 mg, CPB-0.6: 0.6 mg, CPB: 1 mg, CPB-1.2: 1.2 mg. Values represent the mean and standard deviation, $n = 3$ independent samples.

Fig. R16. Potential applications in *in vivo* hemostasis and activity monitoring. **a** Schematic illustration of the hemostatic process using CPB in a rat liver perforation wound model. **b** Images of the hemostatic effect on damaged livers in the blank group and the CPB group. Yellow dotted line: position of the wound or CPB. **c** Bloodstain on the surface of filter paper in the blank group and the CPB group at 120 seconds. **d** Total blood loss in the blank group and the CPB group. Values represent the mean and standard deviation, $n = 3$ independent samples. *, significant difference compared to the blank group, $p < 0.05$. **e** Images of the hemostatic effect of CPB in a rat dynamic heart perforation wound model. Yellow dotted line: the position of the wound or the CPB. **f** Schematic illustration of CPB as a physical activity monitor and the adhesion of the CPB monitor on a nude mouse. **g** Potentials of the nude mouse from under anesthetic to wake up state.

Revision:

[Page 9, Paragraph 1]

“The presence of the BP nanosheets improved the electrical conductivity and sensing sensitivity of the patches (Supplementary Fig. 8). The electrical resistivity of CPB was $\sim 0.35 \text{ ohm} \times \text{cm}$, closely matching that of Cu foil ($\sim 0.33 \text{ ohm} \times \text{cm}$) and significantly lower than that of CP ($\sim 0.70 \text{ ohm} \times \text{cm}$), suggesting good electrical conductivity of CPB...”

[Page 13, Paragraph 2]

“With its conductive properties, CPB can be directly adhered to the skin of a nude mouse and used as a physical activity monitor (Fig. 4f). Fig. 4g illustrates the electromyographic (EMG) signals acquired by CP, CPB and Cu foil of the nude mouse from under anesthetic to wake up state. CPB and Cu foil presented the similar level of sensing capability, which was superior to that of the CP sample. The physiological activities of wake-up processes could be clearly identified through the amplitude and frequency of the sensing potentials. The presence of BP nanosheets improved the electrical conductivity and sensitivity of the patches, allowing CPB to potentially be popularized as a biomedical sensor.”

(3) For the study on tumour recurrence prevention. The patches were adhered to the surgical site after tumour resection, where the enhanced bioadhesion ensured that the physical position of potential tumour cells was fixed to receive NIR light. The experimental results demonstrated that the CPB therapeutic approach possessed effective *in situ* antitumour functions for postsurgical treatment. We also added the information in the section of Potential applications and highlighted in our revised manuscript.

Revision:

[Page 15, Paragraph 1]

“The enhanced bioadhesion ensured that potential tumour cells remain in a fixed position to receive NIR light...”

Q3 There're arbitrary claims in the manuscript that could be adjusted and typos to be corrected. In the Abstract, for example, "useless hydrogen bonds" could be replaced with better terms.

Response:

We thank for the reviewer's suggestion. The sentence has been revised and highlighted in revised manuscript.

Revision:

[Page 2, Abstract]

“...the hydration film on wet tissues can generate a boundary, forming hydrogen bonds with the adhesives that weaken adhesive strength...”.

In addition, the revised manuscript has been thoroughly polished by Springer Nature Author Services with a certification (verification code 1C8D-C677-E47E-4C6E-BAEP).

Reviewers' Comments:

Reviewer #1:

Remarks to the Author:

This revised version is qualified for the publication on Nature Communication.

Reviewer #2:

Remarks to the Author:

Thank you to the authors for addressing the comments.

Reviewer #3:

Remarks to the Author:

All the comments from the reviewer have been nicely addressed. There remains to be typos in the Abstract and the Main Text of the manuscript. For example, "Introducing black 25 phosphorus (BP) is believed to to enhance the water absorption capacity" in the Abstract. The authors should double check their writings carefully. Other than that, I have no further comments.

Response to referees' and editors comments

Editors:

I am delighted to say that we are happy, in principle, to publish it under the open access CC BY license (Creative Commons Attribution 4.0 International License). First, we ask you to revise your paper one last time to address our editorial requests (in the attached author checklist) and any remaining comments from reviewers (included at the end of this email, if applicable).

Response:

We deeply appreciate for your consideration of our paper could be published in principle. We have revised the manuscript according to the reviewers' comments and editorial requests. The detailed point-by-point responses and the checklist are uploaded with this manuscript.

Reviewer #1:

This revised version is qualified for the publication on Nature Communication.

Response:

We thank the reviewer for giving positive comment that our revised version is qualified for the publication on Nature Communication

Reviewer #2:

Thank you to the authors for addressing the comments.

Response:

We thank the reviewer for accepting our response and giving positive comment.

Reviewer #3:

All the comments from the reviewer have been nicely addressed. There remains to be typos in the Abstract and the Main Text of the manuscript. For example, "Introducing

black phosphorus (BP) is believed to to enhance the water absorption capacity" in the Abstract. The authors should double check their writings carefully. Other than that, I have no further comments.

Response:

We thank the reviewer for accepting our response and giving positive comment. We have further checked and correct the writings of the manuscript.